# An iPSC-based in vitro model recapitulates human thymic epithelial development and multi-lineage specification

Yann Pretemer [1], Yuxian Gao[1,2], Kaho Kanai[1,3], Takuya Yamamoto [1,4,5], Kohei Kometani [1], Manami Ozaki[1,3], Karin Nishigishi[1,3], Tadashi Ikeda[6], Huaigeng Xu[1,7], Akitsu Hotta [1] & Yoko Hamazaki [1,2] ✉

Thymic epithelial cells (TEC) are crucial in supporting T cell development, but their high heterogeneity and difficulty of isolation pose obstacles to their study in humans. Particularly, how diverse TEC lineages arise from a common progenitor remains poorly understood. To address this, here we establish a human iPSC-based model of thymus organogenesis capable of deriving these lineages in vitro. Through controlled retinoid signaling followed by self-directed differentiation, we obtain FOXN1+ TEC progenitor-like cells and diverse mature MHCII+ populations resembling cortical and medullary TECs, allowing us to infer their developmental trajectories. Upon thymocyte co-culture, induced TECs support the generation of naïve T cells with diverse TCR repertoires and further develop into *AIRE*+ and mimetic TEC subpopulations. Our system provides a fully in vitro model of human TEC differentiation from early fate specification to late-stage maturation, offering new insights into human thymus development and potential regenerative applications for congenital thymic disorders.

Thymic epithelial cells (TEC) constitute the major stromal cell compartment in the thymus and are crucial in supporting the differentiation and selection of T cells[1–3]. They arise from the ventral side of the third pharyngeal pouch (3rd PP) endoderm, where the thymic anlage forms and starts to express the TEC master regulator *FOXN1* by 6 weeks of gestation in humans[4–6]. As surrounding mesenchymal cells and lymphoid progenitors from the bone marrow start to populate the early thymic anlage, FOXN1+ TEC progenitors (TEP) give rise to cortical TECs (cTEC), which facilitate the lineage commitment and positive selection of thymocytes, and medullary TECs (mTEC), which are responsible for negative selection and regulatory T cell (Treg) generation to ensure self-tolerance[1,4,7]. Recent murine studies have revealed a remarkable heterogeneity of TEC subpopulations crucial for

thymic function, including MHCII^high cells (mTEC^high) variably expressing Aire and peripheral tissue antigens, MHCII^low cells (mTEC^low) secreting Ccl21 and other cytokines for thymocyte chemoattraction, intertypical TECs with both cTEC and mTEC characteristics, and various mimetic mTECs imitating extrathymic cell types, each with distinct molecular characteristics[8–15]. Defects in specific TEC subpopulations lead to impaired T cell development and autoimmunity[16,17].

However, how this stunning diversity of TECs arises during human development remains unclear. Particularly, upstream signals and molecular mechanisms of *FOXN1* leading to TEC fate specification in the 3rd PP are only poorly understood[18], and the developmental processes underlying the establishment of specific lineage identities in

[1]Center for iPS Cell Research and Application, Kyoto University, Kyoto, Japan. [2]Laboratory of Immunobiology, Graduate School of Medicine, Kyoto University, Kyoto, Japan. [3]Department of Human Health Sciences, Graduate School of Medicine, Kyoto University, Kyoto, Japan. [4]Institute for the Advanced Study of Human Biology, Kyoto University, Kyoto, Japan. [5]RIKEN Center for Advanced Intelligence Project, Kyoto, Japan. [6]Department of Cardiovascular Surgery, Graduate School of Medicine, Kyoto University, Kyoto, Japan. [7]Eli and Edythe Broad Center of Regeneration Medicine and Stem Cell Research, University of California, San Francisco, San Francisco, CA, USA. ✉e-mail: yoko.hamazaki@cira.kyoto-u.ac.jp

TEC subpopulations have not been clarified. Since abnormalities in TEC development cause immunodeficiencies and autoimmune diseases with high morbidity and mortality[19–21], a deeper understanding of these processes is of critical importance to improve patient outcomes and find new therapeutic strategies. Existing avenues for research include mouse models, which have hinted at complex mechanisms of shifting TEP identities and differentiation biases during development[22–27], but they are limited by their stark differences in gestation time and thymus morphology compared to humans, such as the near-absence of Hassall's corpuscles[28]. Human primary TECs have also served as important platforms, allowing the identification of various TEC subsets through recent advances in scRNA-seq[29–33], but primary tissues are difficult to obtain from human embryos, contain fewer than 1% TECs, exhibit high sample-to-sample variability[30], and cannot be continuously observed during development. When placed into culture, primary TECs rapidly downregulate their *FOXN1* expression[34,35], preventing the study of their developmental dynamics. To advance our understanding of human thymus development, a model capable of recapitulating this process in vitro would be invaluable.

Human pluripotent stem cells (PSC) have served as a powerful tool to derive cells of all three germ layers and observe their development in vitro[36]. In the past decade, several reports have shown successful induction of *FOXN1*+ TEP-like cells able to contribute to the reconstitution of thymic function in immunodeficient mice upon in vivo transplantation[37–49]. However, in vitro, these cells remain characteristic of immature or intertypical TECs and cannot yet be differentiated into the diverse mature cTEC and mTEC lineages present in the human fetus, limiting their use as a model of thymus organogenesis. Faithful in vitro recapitulation of the full process of TEC development from PSCs would allow the analysis of fate decisions and trajectories that are difficult to observe in vivo, enable the interrogation of lineage-specific pathological mechanisms, and provide a potential source of specific types of TECs for regenerative applications.

Therefore, in this study, we establish a fully in vitro induction system capable of deriving diverse mature TEC-like lineages from human induced pluripotent stem cells (iPSC). By combining retinoic acid (RA)-based endodermal patterning with a period of self-directed differentiation, we obtain highly heterogeneous populations resembling cTECs, mTEC^low, mTEC^high, and mimetic mTECs. Using *FOXN1^mCherry* reporter lines, we observe the association of *FOXN1* with different TEC phenotypes over time and isolate FOXN1+ cells for functional evaluation through thymocyte co-culture, through which we further obtain *AIRE*+ and post-*AIRE* populations. With single cell profiling, we benchmark our induced cells against primary human TECs and determine their lineage trajectories and potential regulators. Together, our system provides a new window into the complex process of thymus organogenesis in an easily reproducible and accessible human model.

## Results

### Induction of TEP-like cells through RA-based endodermal patterning

To recapitulate the development of TECs in vitro, we established a chemically defined protocol that directs iPSCs through each intermediary stage in 2D culture by manipulating important signaling pathways governing lineage fate decisions. We started from a previously published protocol capable of efficiently producing anterior foregut endoderm (AFE)[50], an intermediary stage from which pharyngeal organs, as well as lungs, are known to develop[51]. Although WNT and BMP signaling have been implicated in thymus development[52–54] and are abundantly used in TEP induction protocols[37–49], they are also known to posteriorize AFE into the lung lineage[55–57]. To specifically induce pharyngeal endoderm (PE), we omitted both WNT and BMP and instead used RA, which directly promotes *HOXA3*, a transcription

factor (TF) specifying the positional identity of the 3rd PP from which the thymus arises[58–60] (Fig. 1a). We also included FGF8, which is required for the development of the 3rd PP derivatives[61,62].

By day 3 of the induction using the standard 201B7 iPSC line[63], cells adopted a characteristic endodermal morphology and expressed markers of definitive endoderm (DE), with nearly all cells being EPCAM+CXCR4+ (Fig. 1b–d). Markers of pluripotency were downregulated by day 7, when AFE markers were highly expressed and 70% of cells were SOX2+FOXA2+ (Fig. 1c, d). By day 18, cells were overconfluent and started forming dense patches, with PE markers TBX1 and HOXA3 being widely expressed (Fig. 1e). High expression of TFs crucial for thymus organogenesis[64–68] was seen by day 18, leading to the expression of *FOXN1* at around half of the level of purified human primary TECs from pediatric donors by day 28 (Fig. 1f). Strikingly, it is only a narrow range of RA concentrations from day 7 to 18 that eventually led to the expression of PE and 3rd PP markers, indicating that the precise level of *HOXA3*, which was dose-dependent on RA, is crucial in correctly specifying the 3rd PP and allowing the development of its derivatives (Fig. 1g). This is supported by the concurrent expression of *GCM2*, a marker of the parathyroid, which also arises from the 3rd PP, but on the dorsal side[69]. RA alone was necessary and sufficient for this process, although the addition of FGF8 further increased PE markers and *FOXN1* (Fig. 1h).

Overall, similar results in marker expression and morphology were seen in two more commonly used iPSC lines, 409B2[70] and 1383D6[71,72] (Supplementary Fig. 1a, b). Markers of other endodermal lineages, including thyroid and pancreas, were also detected at low levels on day 28 (Supplementary Fig. 1c). When RA or FGF8 supplementation was extended until day 28, no further increase in *FOXN1* expression was seen compared to no supplementation after day 18, indicating that their effect is temporally restricted to the PE stage (Supplementary Fig. 1d). Similarly, when adding agonists or antagonists of major developmental signaling pathways from day 18 to 28, we found that neither BMP nor WNT stimulation increased *FOXN1* expression (Fig. 1i). On the contrary, high WNT activation abolished *FOXN1* and *GCM2* expression, reflecting the disruption of thymus and parathyroid development in mice with elevated WNT signaling in TECs[73]. NOTCH inhibition increased *GCM2* but not *FOXN1*, suggesting a possible role of this pathway in promoting the parathyroid fate over the thymic fate. Together, these results indicate that endodermal patterning based on RA modulating *HOXA3* is sufficient to drive *FOXN1* expression and induce TEP-like cells after the AFE stage, while the modulation of other signaling is largely dispensable. Indeed, signaling molecules of pathways involved in both early and later stages of TEC development, such as NOTCH[74,75] and NF-kB[76,77], automatically increased over time without further signaling manipulation, suggesting possible self-directed differentiation after 3rd PP specification (Supplementary Fig. 1e).

### Self-directed differentiation of cTEC- and mTEC-like populations

Since *FOXN1*+ TEP-like cells autonomously emerged after the PE stage, we next opted to maintain the induced cells in long-term culture to allow potential differentiation into all TEC lineages. To identify TEC-like cells and monitor their developmental changes over time, we constructed *FOXN1^mCherry* reporters from all three iPSC lines (Supplementary Fig. 2a). All *FOXN1^mCherry* reporter lines had heterozygous insertion of the T2A-mCherry construct immediately after the last codon of the *FOXN1* gene, showing no indels in the non-targeted allele and maintaining the normal karyotype of their parent lines (Supplementary Fig. 2b, d, e). Cell morphology was normal at the iPSC stage, and the expression of PE markers and *FOXN1* was similar to the parent lines on day 28 (Supplementary Fig. 2c, f). Consistent with the mRNA expression results (Fig. 1f and Supplementary Fig. 2f), mCherry fluorescence was visible by day 28, followed by an increase in brightness over time (Fig. 2a). Notably, while only a single mCherry+ population

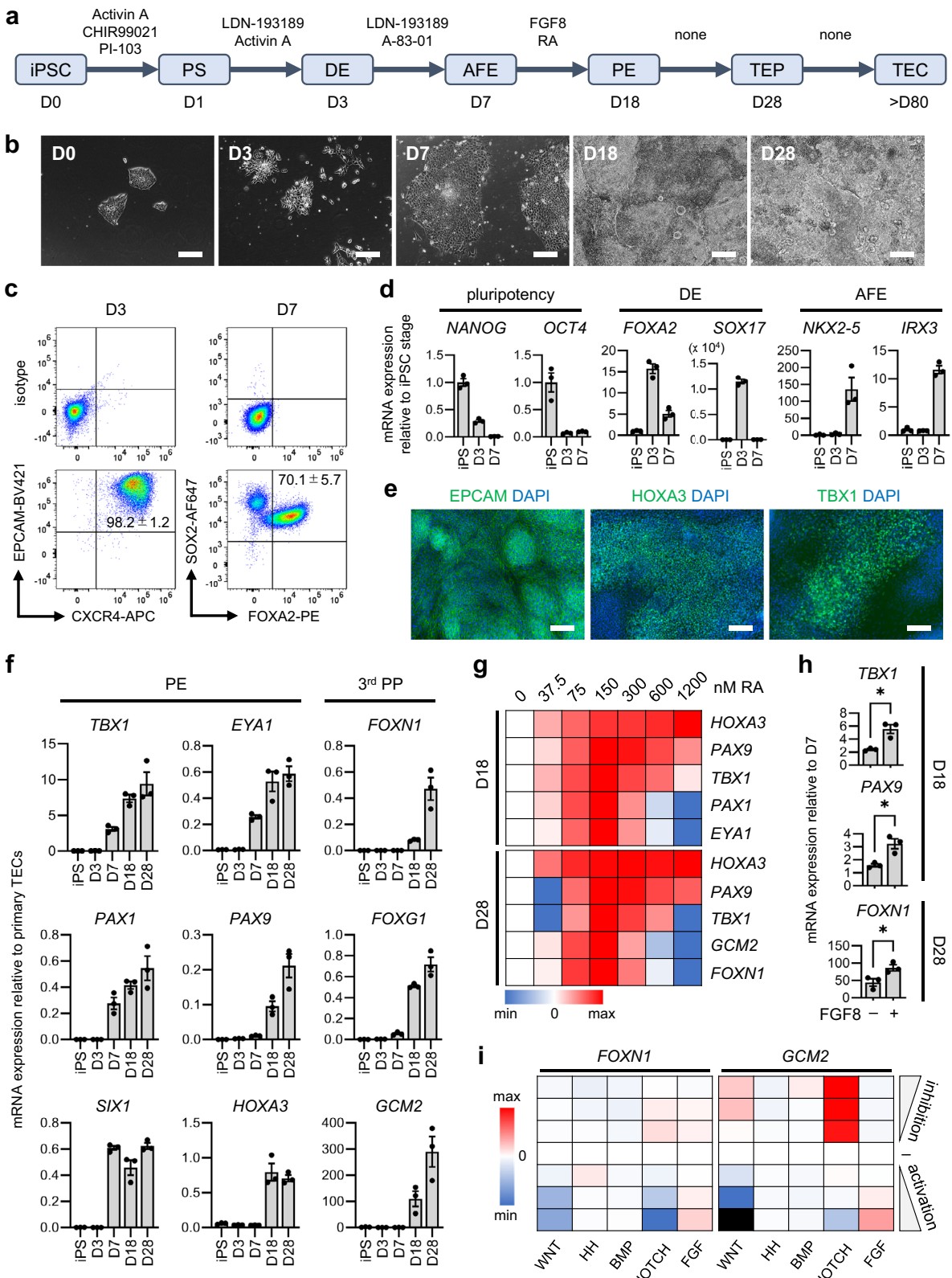

was present on day 28, distinct mCherry^high and mCherry^low populations were evident by day 57, both of which were still maintained on day 133, indicating self-directed differentiation into stable, heterogeneous TEC-like lineages (Fig. 2b–d). By day 133, 3D-like nodules up to 100 μm in height were observed within mCherry^+ colonies (Supplementary Video 1).

When isolating the mCherry^high, mCherry^low, and mCherry^- populations, we found that cTEC-specific genes such as *PSMB11* and *PRSS16*, involved in self-peptide generation[16,78,79], were exclusively expressed by mCherry^high cells at levels matching or exceeding human primary TECs by day 39 (Fig. 2e). In contrast, expression of mTEC-specific genes such as *CCL19* and *CCL21*, crucial for thymocyte chemoattraction

**Fig. 1 | TEC induction from human iPSCs in a 2D in vitro system. a** Schematic of the TEC induction protocol. D, day; iPSC, induced pluripotent stem cell; PS, primitive streak; DE, definitive endoderm; AFE, anterior foregut endoderm; PE, pharyngeal endoderm; TEP, thymic epithelial progenitor; TEC, thymic epithelial cell. **b** Phase-contrast images of induced cells at each time point. Scale bars, 200 μm. **c** Representative flow cytometry results (bottom) with the respective isotype controls (top) on D3 and D7. **d** mRNA expression of pluripotency, DE, and AFE markers over time. **e** Representative images of immunostaining on D18. Scale bars, 100 μm. **f** mRNA expression of PE and third pharyngeal pouch (3ʳᵈ PP) markers over time. **g** Heatmap of the mean $\log_2$ fold change in mRNA expression in response to retinoic acid (RA) addition from D7 to D18 compared to DMSO only. **h** mRNA expression with or without addition of 50 ng/ml FGF8 from D7 to D18. **i** Heatmap of the mean $\log_2$ fold change in mRNA expression in response to addition of inhibitors or activators of specific signaling pathways from D18 to D28 compared to untreated

(-) controls on D28. Black fields represent cases where expression was undetected in all experiments. The signaling modulators and concentrations are as follows: WNT inhibition, 0.2 μM, 1 μM, 5 μM XAV939; WNT activation, 0.4 μM, 2 μM, 10 μM CHIR99021; HH inhibition, 0.2 μM, 1 μM, 5 μM Cyclopamine; HH activation, 20 nM, 100 nM, 500 nM SAG; BMP inhibition, 40 nM, 200 nM, 1 μM LDN-193189; BMP activation, 4 ng/ml, 20 ng/ml, 100 ng/ml BMP4; NOTCH inhibition, 0.1 μM, 0.5 μM, 2.5 μM RO4929097; NOTCH activation, 0.2 mM, 1 mM, 5 mM Valproic acid; FGF inhibition, 40 nM, 200 nM, 1 μM BGJ398; FGF activation, 2 ng/ml, 10 ng/ml, 50 ng/ml FGF8. All results are from $n = 3$ independent experiments using 201B7 and values indicate the mean ± SEM (standard error of the mean). Statistical significance was determined using unpaired two-sided $t$-tests (n.s. no significant difference, $*p < 0.05$). Exact $p$-values in (**h**) are as follows: *TBX1*, 0.0106; *PAX9*, 0.0146; *FOXN1*, 0.0374.

---

to the medulla[80,81], was found preferentially in mCherry^low cells or dispersed throughout all populations at levels similar to or lower than primary TECs by day 133. Only *AIRE* was not expressed, probably due to the lack of thymocyte-derived signals[82,83]. These results suggest a preponderance of cTEC-like phenotypes within mCherry^high and early mTEC-like phenotypes within mCherry^low, consistent with previous reports showing higher *FOXN1* expression in cTECs compared to mTECs[35,84]. Similar results were observed at the protein level on day 80, with cTEC marker PSMB11 being seen exclusively within bright mCherry^+ areas, likely corresponding to the mCherry^high population in flow cytometry, and mTEC marker KRT5 being seen preferentially in areas with little to no mCherry fluorescence (Fig. 2f). KRT8 and CLDN4, used in murine studies to identify cTECs[85] and mTECs[86] respectively, were seen throughout the culture similar to epithelial marker EPCAM, suggesting that they may already be expressed in mCherry^- progenitors or possibly in cells of other epithelial lineages (Supplementary Fig. 3a). By day 133, PSMB11^+ cTEC-like areas and KRT5^+ mTEC-like areas were largely spatially separated with some overlapping junctional areas where 3D-like nodules formed, implying the presence of distinct niches similar to the thymus[33,76,87] (Fig. 2g and Supplementary Video 2).

Over time, expression of several PE markers remained high in the mCherry^high population but decreased to levels similar to primary TECs in the mCherry^low population (Supplementary Fig. 3b). Of note, after a period of co-expression with *FOXN1* on day 28 to 39, *GCM2* and *TBX1* starkly decreased in mCherry^high and mCherry^low cells (Supplementary Fig. 3b). Since *GCM2* and *TBX1* are essential for parathyroid organogenesis[88], this suggests the presence of a transcriptional program suppressing the parathyroid fate once *FOXN1* is expressed. Both NOTCH and NF-kB signaling genes remained highly expressed after day 28 in all populations, with NF-kB being especially prominent in the mCherry^high population, and postnatal TEC progenitor marker[29,30] *DLK2* being expressed by mCherry^high and mCherry^low cells in the later stages (Supplementary Fig. 3c). Ki-67^+ cells were detected even on day 133, suggesting the development and long-term maintenance of a progenitor-like population in our system (Supplementary Fig. 3d).

**Induced TECs are mature and capable of positive selection in vitro**

Importantly, after day 28, our induced TECs (iTEC) started expressing functional markers, such as *IL7* and *DLL4*, crucial for the development of T cells in the thymus[89,90] (Fig. 3a). MHC class II molecule HLA-DR, required for the selection of CD4^+ T cells, was seen especially in clusters in the central parts of mCherry^+ colonies, indicating the emergence of functionally mature iTECs from these areas (Fig. 3b). HLA-DR and HLA-DQ were seen predominantly in the mCherry^high population, where the majority of the cells also expressed CD205, a marker of cTECs[91], although some expression also appeared in mCherry^low and mCherry^- cells by day 133 (Fig. 3c–e). These results suggest that most of the mature iTECs in our system are characteristic of cTECs, while

mTEC-like cells take longer to develop and mature, consistent with previous research indicating cTECs maturing earlier than mTECs in the human fetus[32]. MHC class I molecules HLA-A, -B, and -C, required for the selection of CD8^+ T cells, were maintained in all populations. CD90, a mesenchymal marker recently shown to also be expressed in TECs, possibly contributing to hybrid epithelial-mesenchymal characteristics[35], was also detected in most mCherry^+ cells. Overall, similar results were seen in another 201B7 *FOXN1^mCherry* reporter clone, as well as *FOXN1^mCherry* reporters established from 409B2 and 1383D6 iPSC lines (Supplementary Fig. 4a–g and 5a–e).

We then proceeded to assess the functionality of our iTECs. Up to now, various systems for in vitro T cell production have been developed, including DLL1-expressing stromal cell lines and feeder-free DLL4 microbead systems, but while they are highly adept at generating CD4^+CD8^+ double-positive (DP) T lineage cells, efficient induction of mature single-positive (SP) cells, especially CD4^+ SP cells, remains challenging[41,92–94]. Therefore, we tested the capability of our iTECs to engage in positive selection, a crucial event in the development of SP cells orchestrated by cTECs in the thymus[1,2,7]. We sorted day 37 mCherry^+ iTECs and co-cultured them with pre-selection (CD69^-) DP thymocytes from pediatric human thymi in 3D organoids at the air-liquid interface, similarly to a previous report using the murine stromal MS5-hDLL1 cell line[92], with the same cell line used as control (Fig. 4a, b and Supplementary Fig. 6a, b). After 2 weeks of organoid co-culture, MS5-hDLL1 still had 91% of the CD3^+TCRαβ^+ population remaining as CD4^+CD8^+ DP, consistent with the previously reported long co-culture periods required to produce mature SP cells[92], while iTECs led to CD4^+ and CD8^+ SP phenotypes in respectively 47% and 21% of CD3^+TCRαβ^+ (Fig. 4c, d). Within both CD4^+ and CD8^+ SP cells, the percentage of CD62L^+CD45RA^+CCR7^+ was significantly higher for iTECs compared to MS5-hDLL1, consistent with the acquisition of a mature naïve T cell phenotype. Furthermore, 55% of the total CD8^+ SP cells resulting from iTEC co-culture were CD8αβ^+, while only 22% were CD8αβ^+ in the case of MS5-hDLL1, indicating that agonist selection resulting in innate-like CD8αα phenotypes[95] was dominant in MS5-hDLL1, while high rates of positive selection were achieved by iTECs. Unlike MS5-hDLL1, iTECs were also capable of generating 8% TCRγδ^+ cells, similarly to the human thymus where some TCRγδ^+ cells develop along the DP pathway[96]. To confirm the positive selection of CD4^+ and CD8^+ SP cells with diverse TCR rearrangements by iTECs, we performed TCR repertoire analysis after the 2 weeks of co-culture. TCRβ repertoires of both CD4^+ and CD8^+ SP cells generated through iTECs showed high diversity similar to that of pediatric donor-derived CD4^+ and CD8^+ SP thymocytes, which had comparable gene usage frequencies and Shannon index[97] values (Fig. 4e and Supplementary Fig. 6c, d). When co-culturing mCherry^+ iTECs with earlier-stage CD4^-CD8^- double-negative (DN) thymocytes, we found a greater propensity toward innate-like TCRγδ and CD8αα phenotypes, likely in part due to the inherent heterogeneity of DN cells[98], but conventional CD3^+TCRαβ^+

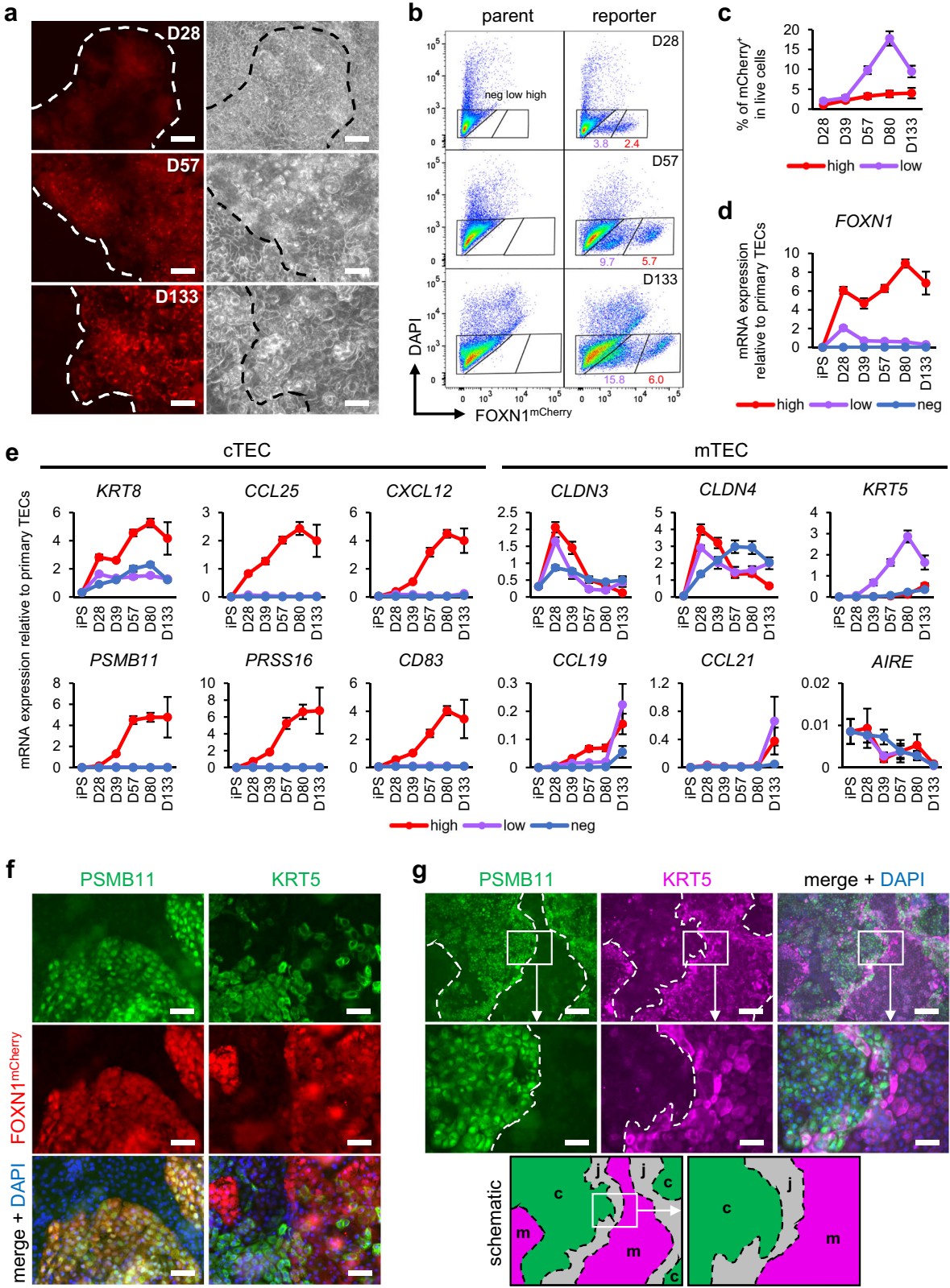

thymocytes including 45% DP, 8% CD4⁺ SP, and 42% CD8⁺ SP cells were also generated (Supplementary Fig. 7a–d). In contrast, mCherry⁻ iTECs could neither induce SP thymocytes nor maintain DP or DN thymocytes in co-culture (Supplementary Fig. 6e, 7e). Together, these results demonstrate the capability of our iTECs, specifically the mCherry⁺ population, to support both early T cell development and later positive selection of SP cells with diverse TCR repertoires.

## iTECs consist of heterogeneous populations resembling primary TECs

Next, to characterize the different induced lineages and trace their development, we performed scRNA-seq at multiple time points during the induction. The resulting dataset contained a major *EPCAM*-expressing epithelial partition, as well as minor non-epithelial clusters including neurons, muscle, and mesenchyme, with similar cell

**Fig. 2 | Differentiation of cTEC- and mTEC-like cells over time. a** FOXN1^mCherry fluorescence (left) and phase-contrast (right) images over time. Scale bars, 50 µm. D, day. Representative of $n = 6$ (D28 and D57) or $n = 4$ (D133) independent experiments. **b** Representative flow cytometry results of the reporter compared to the parent line at each time point. **c** Quantification of the percentage of mCherry^high and mCherry^low cells using flow cytometry. All values indicate the mean ± SEM from $n = 6$ (D0 to D80) or $n = 4$ (D133) independent experiments. **d, e** mRNA expression of thymic epithelial cell (TEC) markers in the mCherry^high, mCherry^low, and mCherry^- (neg) populations over time. All values indicate the mean ± SEM from $n = 6$ (D0 to

D80) or $n = 3$ (D133) independent experiments. cTEC, cortical TEC; mTEC, medullary TEC. **f** Representative images of immunostaining on D80 from $n = 6$ independent experiments. Scale bars, 50 µm. **g** Representative images of immunostaining on D133 from $n = 3$ independent experiments. Scale bars, 200 µm (top row) and 50 µm (middle row). The bottom row schematically depicts the PSMB11^+ area (green), the KRT5^+ area (magenta), and the overlapping junctional area (gray) as seen in the immunostaining. c, cTEC-like; m, mTEC-like; j, junction. All results were obtained using the 201B7 FOXN1^mCherry reporter iPSC line.

distributions in two separate inductions (Fig. 5a–c and Supplementary Fig. 8a, b). Epithelial cells, starting from a common AFE progenitor expressing *OTX2* on day 7, moved through the PE stage and diverged into several clusters with varying levels of *FOXN1* expression (Fig. 5a–c). From these terminal clusters, we identified one cTEC-like cluster and five mTEC-like clusters by scoring each cell according to its expression of the top human primary cTEC and mTEC markers defined by Park et al. [30] (Fig. 5d). As expected, *FOXN1* expression was highest in the cTEC cluster, which also exclusively showed high expression of cTEC markers including *PSMB11, LY75, CCL25, DLL4*, etc. (Fig. 5e, f). In contrast, *FOXN1* expression was generally low in the five mTEC clusters, which showed heterogeneous expression of diverse mTEC markers including *CLDN4, CCL19, KRT5, CD24*, etc. Mature TEC markers *HLA-DRA* and *CD74* were observed in the cTEC, mTEC I, and mTEC II clusters. Interestingly, a small cluster scoring high for mTEC marker expression while simultaneously expressing *KRT1* and *KRT10*, characteristic of keratinized cells (KC) present in Hassall's corpuscles[99–102], was also detected (Fig. 5d, e). The top differentially expressed genes in each of these terminal clusters constitute both novel and previously defined[30,80,86,103–107] markers of TECs, including putative markers from recent scRNA-seq studies[29,30,108,109] such as *IGFBP6, KRT15, KRT17, GAS6, AQP3, GNG11, PLTP, ELF3*, and *ANXA1*, confirming the identity of our iTECs and their use in finding and validating new markers (Fig. 5g).

To ascertain that the findings of our iTECs are biologically relevant and reflective of actual TECs in the human body, we isolated thymic stroma from three pediatric donors and performed scRNA-seq for comparison. Apart from mesenchyme, endothelium, and other non-epithelial cells, five TEC clusters expressing *EPCAM* were identified (Fig. 6a, b, and Supplementary Fig. 8c). Similarly to iTECs, primary cTECs uniformly expressed high levels of *FOXN1, PSMB11*, and *CCL25*, while primary mTECs were highly diverse and variably expressed *CCL19, CLDN4*, and *CD24* (Fig. 6b, c). The mTEC cluster labeled mTEC-CCL19/21 was marked by high expression of cytokines *CCL19* and *CCL21* and keratins *KRT5* and *KRT15*, with lower expression of *HLA-DRA*, similar to previously defined mTEC^low populations[8,9,29]. The mTEC-CLDN3/4 cluster showed high expression of different claudins, such as *CLDN3* and *CLDN4*, and *HLA-DRA*, characteristic of previously defined mTEC^high populations[8,29,86]. This cluster also contained the *AIRE*^+ subpopulation, as well as cells expressing KC marker *KRT10* or secretory cell marker *SLPI*, indicating the presence of mimetic mTECs[15]. However, most mimetic mTECs were found in the cluster labeled mTEC-mim, which included diverse populations expressing neuronal marker *NEUROD1*, myocyte marker *MYOG*, and many others. In addition to these diverse mTECs, TECs with an intertypical phenotype labeled mcTEC were also detected, and these cells expressed various keratins and other markers of progenitor-like cells[30,31].

To directly compare the terminal clusters of iTECs to primary TECs, we integrated the samples of the primary thymic stroma with the iTEC samples from day 133 (Fig. 6d). Cells from the induced cTEC cluster co-clustered with primary cTECs and bridged the gap between cTECs and mcTECs, indicating they consist of a mixture of cTEC-fated progenitor-like cells and mature cTECs indistinguishable from their primary counterparts. Induced mTEC I cells co-clustered predominantly with the primary mTEC-CCL19/21 cluster, indicating an

mTEC^low-like phenotype, with some cells also overlapping with mcTECs. The induced mTEC II cluster showed substantial overlap with primary mTEC-CLDN3/4, suggesting an mTEC^high-like phenotype, while also extending toward mTEC-mim. Both induced mTEC III and KC clusters co-clustered with mTEC-CLDN3/4, with the former showing near-complete overlap with the subpopulation expressing secretory cell markers, and the latter co-clustering closely with the keratinized subpopulation (Fig. 6e, f). Multiple markers of secretory cells were indeed highly expressed in the induced mTEC III population, including *SLPI, CXCL17*, and various mucins and uroplakins reminiscent of secretory mimetic mTECs, indicating the possibility of mimetic mTEC-like cells being induced in our system despite the lack of *AIRE* expression (Fig. 6g). Recent studies have shown that mimetic mTECs, being fewer but not absent in *Aire*^-/- mice, require specific lineage-defining TFs in their development[10,11]. To examine whether such TF activity is recapitulated in iTECs, we performed gene regulatory network (GRN) analysis and identified regulons—groups of genes that consist of a TF and its direct-binding targets—that were highly active in each cluster. Strikingly, among the top regulons in induced mTEC clusters, particularly mTEC III, were *ELF3, GRHL1*, and *GRHL3*, which have been implicated in the development of mimetic mTECs and the regulation of *Foxn1* in murine studies[18,108,109] (Fig. 6h). Consistent with this, expression of *ELF3* and *EHF*, both putative regulators of late mTEC development[109], was especially high in induced mTEC III, suggesting crucial aspects of *AIRE*-independent mimetic mTEC differentiation were recapitulated in our system (Supplementary Fig. 8d). Some induced cells classified as neurons, myocytes, and ciliated cells also co-clustered with the primary mTEC-mim and mTEC-CLDN3/4 clusters, implying they could be mimetic mTEC-like cells, although only the Cil cluster showed high expression of *ELF3* and *EHF* (Supplementary Fig. 8d–f). In contrast, the induced mTEC IV and mTEC V clusters did not co-cluster with primary TECs (Fig. 6d). This is despite their expression of genes indicative of mcTECs or early mTECs, including *BCAM, DLK2, IGFBP6, ITGA6, CXCL14, WNT4, LAMB3*, and various keratins[29–31,110,111], but no thyroid, pancreas, lung or other epithelial lineage markers, indicating they could be either a transient or aberrant population not seen in vivo (Supplementary Fig. 8g).

Interestingly, the induced Mes cluster partially co-clustered with primary thymic mesenchyme (Fig. 6d), raising the question of whether these cells could affect iTECs similarly to how mesenchyme shapes the thymic microenvironment by secreting growth factors and extracellular matrix (ECM) components[112]. Therefore, we performed ligand-receptor interaction analysis based on the expression of potential ligands on the Mes cluster and potential receptors on AFE-derived terminal clusters on day 133 (Supplementary Fig. 9a). This revealed high expression of BMPs, FGFs, and ECM components such as collagens and laminins in the Mes cluster, with appropriate receptors being heterogeneously expressed in iTEC clusters, implying the recapitulation of a thymus-like microenvironment with epithelial-mesenchymal interactions that may contribute to the self-directed differentiation in our system. These mesenchyme-like cells, marked by VIM, are present by day 28 and closely associate with mCherry^+ colonies by day 133, with localized accumulations of these cells below the mCherry^+ layer being observed in 3D-like nodules (Supplementary

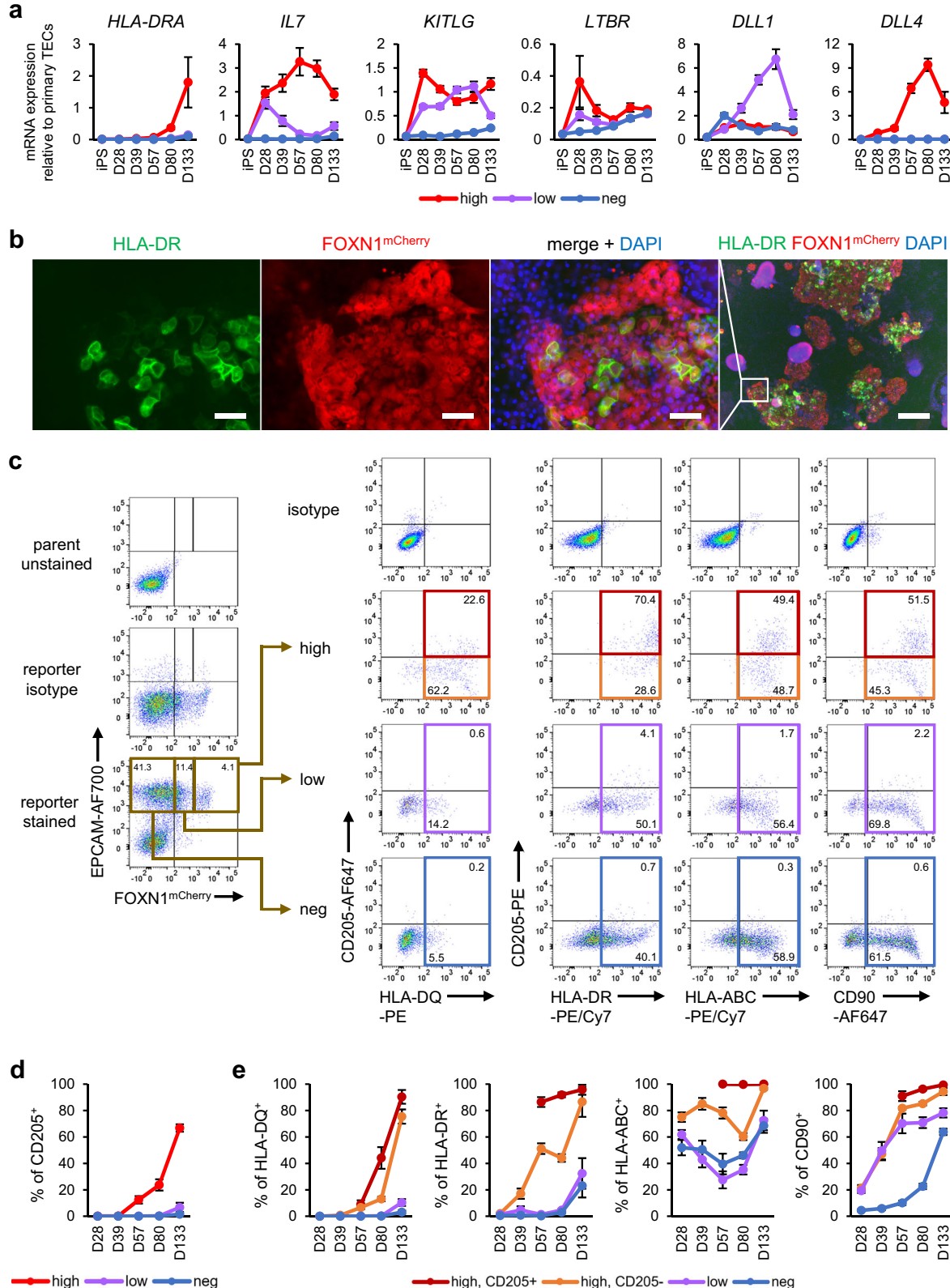

**Fig. 3 | iTECs express functional and mature TEC markers. a** mRNA expression in the mCherry[high], mCherry[low], and mCherry[-] (neg) populations over time. D, day. **b** Representative images of immunostaining on D80 from $n = 6$ independent experiments. Scale bars, 50 μm (left three images) and 500 μm (right image). **c**–**e** Representative flow cytometry results on D133 (**c**) with quantification of CD205[+] cells in each of the EPCAM[+]-gated mCherry[high], mCherry[low], and mCherry[-] populations (**d**) and quantification of HLA-DQ[+], HLA-DR[+], HLA-ABC[+], and CD90[+] cells in the mCherry[high]CD205[+], mCherry[high]CD205[-], mCherry[low], and mCherry[-] populations (**e**). In (**c**), colored gates correspond to the populations used for quantification in (**e**). All results were obtained using the 201B7 *FOXN1[mCherry]* reporter iPSC line. All values indicate the mean ± SEM from $n = 6$ (D0 to D80) or $n = 3$ (D133) independent experiments. TEC, thymic epithelial cell; iTEC, induced TEC.

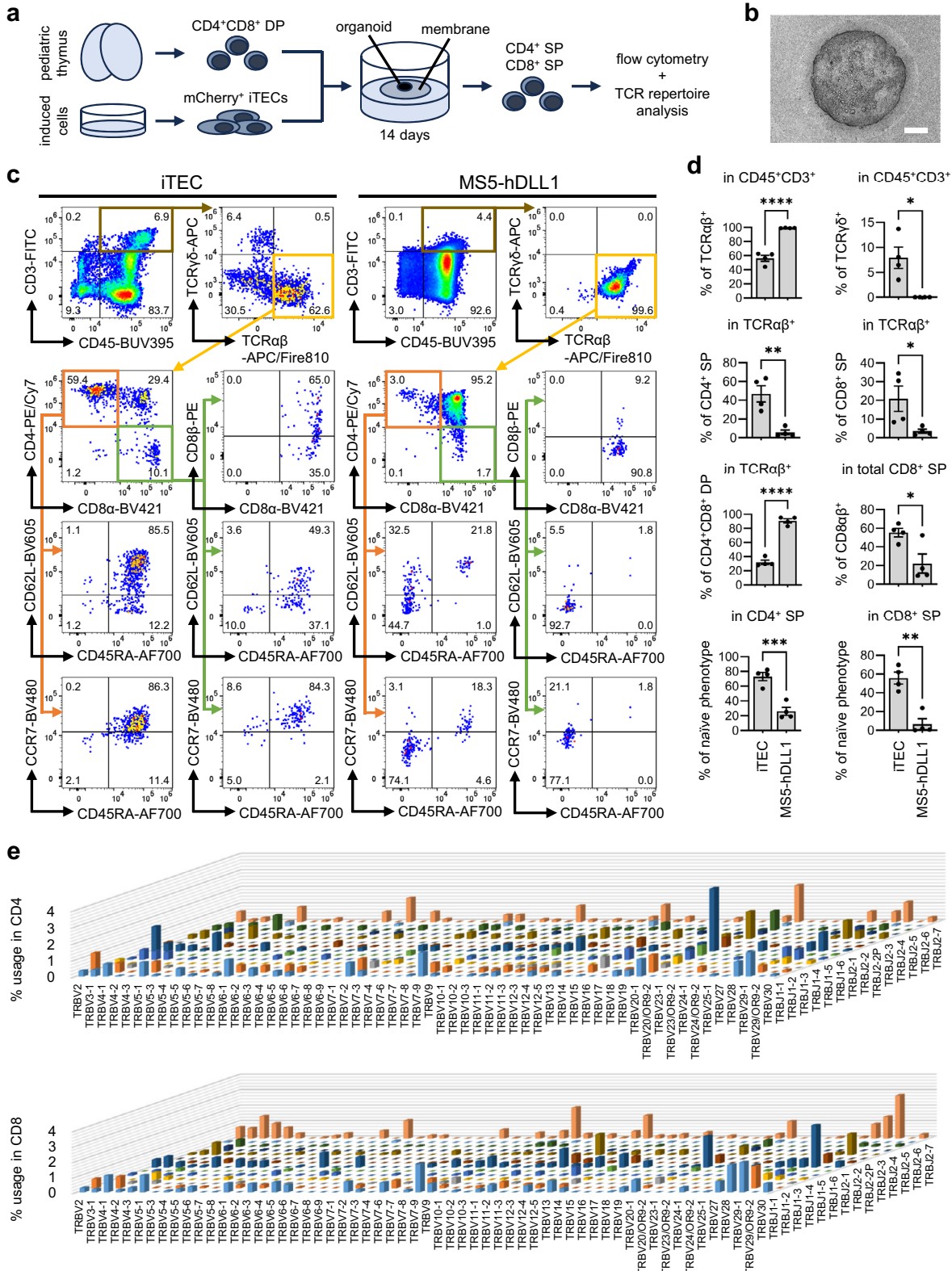

Fig. 9b, c and Supplementary Video 3). We are unable to determine the origin of the Mes cluster due to its lack of connections to other clusters, but iTECs themselves variably express low levels of mesenchymal markers, including *THY1*, *COL1A1*, *VIM*, *FN1*, and *CD44* (Supplementary Fig. 9d), similarly to primary TECs[35], possibly allowing epithelial-to-mesenchymal transition (EMT) to occur[110,113].

## mTEC-like lineages diverge at varying stages of cTEC development

Aiming to shed light on the origins of TEC heterogeneity, we used pseudotime and RNA velocity analysis to determine how the different TEC-like lineages arise from a common progenitor in our system. We find the earliest lineage bifurcation at the PE stage, with one path

**Fig. 4 | Efficient in vitro production of CD4$^+$ and CD8$^+$ T cells with diverse TCR repertoires through iTEC co-culture. a** Schematic of induced thymic epithelial cell (iTEC) co-culture with double-positive (DP) thymocytes. SP, single-positive. **b** Phase-contrast image of an iTEC/thymocyte organoid during co-culture, representative of $n = 4$ independent experiments. Scale bar, 500 μm. **c** Representative flow cytometry results after 14 days of mCherry$^+$ iTEC or MS5-hDLL1 organoid co-culture with DP thymocytes. **d** Quantification of co-culture results from (**c**), with values indicating the mean ± SEM from $n = 4$ independent experiments. Naïve phenotype is defined as CD62L$^+$CD45RA$^+$CCR7$^+$ triple-positive. Statistical significance was determined using unpaired two-sided *t*-tests (n.s. no significant difference, *$p < 0.05$, **$p < 0.01$, ***$p < 0.001$, ****$p < 0.0001$). Exact *p*-values are as follows: TCRαβ$^+$, 0.0000557; TCRγδ$^+$, 0.0102; CD4$^+$ SP, 0.00378; CD8$^+$ SP, 0.0447; CD4$^+$CD8$^+$ DP, 0.00000877; CD8αβ$^+$, 0.0257; naïve phenotype (in CD4$^+$ SP), 0.000944; naïve phenotype (in CD8$^+$ SP), 0.00123. **e** Gene usage of *TRBV* and *TRBJ* in CD4$^+$ (top) and CD8$^+$ (bottom) SP thymocytes obtained after 14 days co-culture of mCherry$^+$ iTECs and DP thymocytes in $n = 1$ independent experiment. All iTEC results were obtained using the 201B7 *FOXN1$^{mCherry}$* reporter iPSC line.

leading to cTEC, mTEC I, and mTEC II clusters, and the other to mTEC III, mTEC IV, mTEC V, and KC clusters, with some possible plasticity between mTEC II and mTEC III (Fig. 7a, b). The TEP cluster, through which the cTEC, mTEC I, and mTEC II trajectories pass, appears by day 28, greatly shifts over time, and disappears by day 133 (Supplementary Fig. 10a). It can be divided into several subclusters, with clusters representing days 28 and 38 already expressing *FOXN1* and some cTEC markers, and clusters dominant at later time points co-expressing cTEC and mTEC markers to varying degrees (Supplementary Fig. 10b, c). Similar shifts in progenitor identity have also been recently reported in mice[27]. Although it did not survive beyond day 49, a small parathyroid cluster was also detected on day 28, indicating that this time point likely corresponds to the point of divergence of the thymic and parathyroid primordia around week 6 of gestation in humans[4,114].

For the cTEC-fated cells, *FOXN1* expression increased and stayed high throughout the induction, whereas mTEC I- and mTEC II-fated cells downregulated their *FOXN1* expression at different points (Fig. 7c). cTEC cytokines *CCL25* and *CXCL12* were expressed by the mTEC I and mTEC II lineages during their progression through the TEP stage and later lost, reflecting murine studies showing at least some mTEC lineages being derived from progenitors with characteristics of early cTECs[22–24]. mTEC I-fated cells also transiently expressed the cTEC-specific proteasome subunit *PSMB11*, while high expression of the cytokine *CCL21*, required for the mTEC function of attracting positively selected thymocytes to the medulla[81], only arose once *PSMB11* expression was lost. Conversely, mTEC cytokine *CCL19* was also expressed to some degree in cTEC-fated cells before being downregulated. Surprisingly, many genes commonly used to identify mTECs, including *CLDN4*[26,86,115,116] and *CD24*[117], were already expressed by the AFE cluster on day 7, suggesting that they may already be present in progenitors before mTEC lineage commitment. In several mTEC-like lineages, expression of secretory or keratinized markers was preceded by the upregulation of *EHF*, *ELF3*, *ELF5*, and *GRHL3*, substantiating the possible role of these TFs in mTEC development[108,109] (Supplementary Fig. 10d).

Genes that were most differentially expressed along pseudotime included many previously identified genes essential for TEC development, including *SIX1*[65] and *PAX1*[64] (Fig. 7d). Cells of the cTEC-like lineage also maintained high expression of *NFKB1*, as well as *GAS6*, which is mediated by NF-kB signaling[118,119], both of which were only transiently seen in mTEC I. *PIK3R1*, *GOLM1*, and *CXXC5*, reported to be highly expressed in cTECs[12,30], were also exclusively expressed along the path leading to cTEC, mTEC I, and mTEC II. In contrast, on the path leading to mTEC III and mTEC IV, high transient expression of *SOX2*, *FOXE1*, and *FOXA1* was observed (Fig. 7d, e). *FOXA1* has been implicated in the development of secretory cells as well as mimetic mTECs[11,120], and a *Foxn1$^-$Sox2$^+$Foxe1$^+$* population has been reported in murine thymi at E11.5[12]. Since expression of these genes decreased by day 133 and no cluster clearly co-expressing them was found in our pediatric thymic stroma, this may constitute an early, transient population fated toward a secretory phenotype (Fig. 7d, e, and Supplementary Fig. 10d, e). The transient expression of *FOXA1* and *FOXE1*, as well as the subsequent upregulation of secretory markers *PSCA* and *CXCL17*, was found almost exclusively in the mCherry$^-$ population, suggesting that this lineage may have the potential to develop independently of *FOXN1*, although this will require further confirmation by lineage tracing (Supplementary Fig. 10f). To examine the signaling that could potentially govern this lineage fate decision, we looked at regulon activity and analyzed differentially expressed genes along pseudotime in the mTEC III compared to the mTEC II lineage. *SOX2*, *FOXE1*, and *FOXA1* regulons were indeed highly active along the path to mTEC III, but not mTEC II, where regulons including *SIX1*, *GATA3*, and *FOXG1* were active (Supplementary Fig. 10g). *SHH*, reported to be upstream of *FOXE1*[121,122], and NOTCH inhibitor *DLK1* were transiently highly expressed in mTEC III-fated cells, whereas NOTCH target gene *HEY1*, as well as *BMP4* and *ID* genes downstream of BMP signaling[123,124] were highly expressed in mTEC II-fated cells (Supplementary Fig. 10h). Furthermore, differential regulon activity between clusters was observed in RA receptors, indicating lineage-specific temporal changes in retinoid signaling (Supplementary Fig. 10i).

Thus, early differential signaling in PE may already determine the lineage fate of some types of mimetic mTECs, concurrently with or even before the establishment of the thymic and parathyroid primordia, while other heterogeneous mTEC populations diverge thereafter from *FOXN1$^+$* progenitors with characteristics of cTECs (Fig. 7f).

## iTECs model *AIRE$^+$* mTEC differentiation in thymocyte co-culture

Since *AIRE* was not expressed at any time point during the induction, we next examined whether iTECs have the potential for *AIRE$^+$* mTEC differentiation by co-culturing them with thymocytes, known to promote mTEC development[82,83]. After 2 weeks of organoid co-culture of day 37 mCherry$^+$ iTECs with DP thymocytes, *AIRE* was significantly upregulated in the mCherry$^+$ fraction, reaching 6% of the expression level of primary TECs (Fig. 8a, b). scRNA-seq of the organoids revealed diverse TEC-like populations similar to the non-co-cultured iTECs, with clusters of cTEC, mTEC I, mTEC II, and mTEC III that co-cluster closely with their primary counterparts (Fig. 8c, d, and Supplementary Fig. 11a, b). Mesenchyme-like clusters were also detected with partial co-clustering of primary mcTECs and induced mTEC I from day 133, as well as their respective Mes clusters, implying that mCherry$^+$ iTECs may partially undergo EMT in organoid co-culture, similarly to TECs acquiring EMT signatures during aging in vivo[125,126] (Fig. 8c, d, and Supplementary Fig. 11a, c). In addition, small lymphocyte clusters including one containing *FOXP3$^+$IL2RA$^+$* cells were present, indicating possible iTEC functionality of inducing Tregs from the DP stage (Fig. 8c, e and Supplementary Fig. 11c). *AIRE* expression was observed within a section of the mTEC II cluster directly connected to clusters co-expressing neuronal markers and *EPCAM*, reminiscent of neuronal mimetic mTECs (Fig. 8e, f). These clusters, assigned as mTEC-mim I-III, co-clustered closely with the mTEC-mim cluster in our dataset of human primary thymic stroma, as well as the TEC(neuro) cluster in the dataset of human thymic epithelium by Park et al.[30] (Fig. 8d and Supplementary Fig. 11b).

RNA velocity analysis, with the inferred latent time representing the cell's internal clock[127], indicates the likely origin of these neuronal

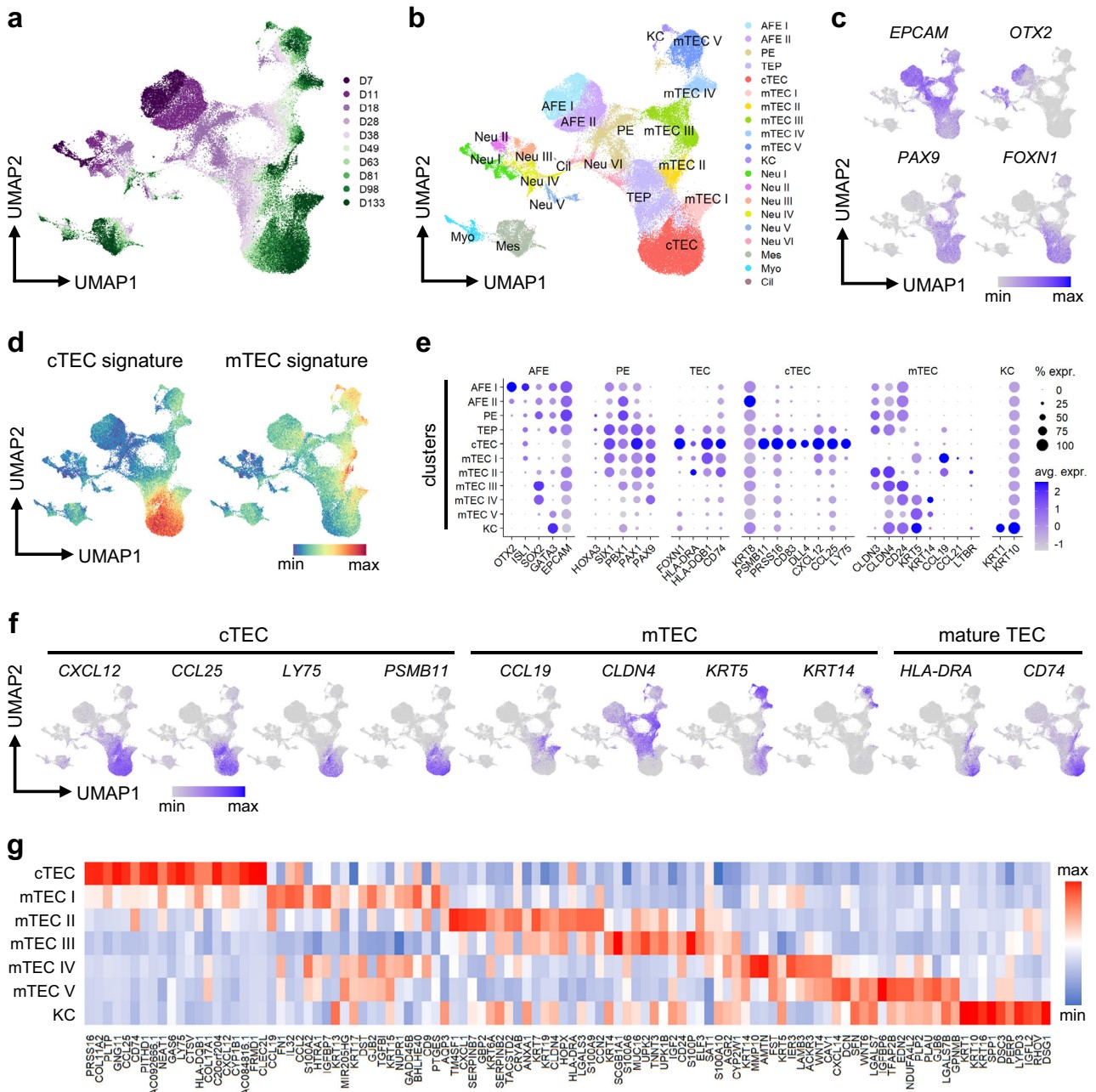

**Fig. 5 | Single cell profiling of iTECs reveals heterogeneous TEC-like populations. a, b** Induced cells from D7 to D133 according to time point (**a**) and unsupervised clustering (**b**) on UMAP. D, day; AFE, anterior foregut endoderm; PE, pharyngeal endoderm; TEP, thymic epithelial progenitor; TEC, thymic epithelial cell; cTEC, cortical TEC; mTEC, medullary TEC; KC, keratinized cell; Neu, neuron; Myo, myocyte; Mes, mesenchyme; Cil, ciliated. **c** Expression of AFE, PE, and TEC marker genes on UMAP. **d** Module scores of the top 20 cTEC and mTEC marker genes as defined by Park et al.[30]. **e** Dot plot of TEC and other marker gene expression in induced AFE-derived lineages. % expr., percentage of cells in which the gene is expressed; avg. expr., average expression. **f** Expression of cTEC, mTEC, and mature TEC marker genes on UMAP. **g** Heatmap of the average expression of the top 20 differentially expressed marker genes in each AFE-derived terminal cluster. All results were obtained using the 201B7 *FOXN1^mCherry* reporter iPSC line.

mimetic mTEC-like cells to be a subset within the mTEC II cluster expressing various keratins (Fig. 8g, h, and Supplementary Fig. 11d–f). The expression of keratins and *BCAM*, reflective of progenitors[31], decreased across pseudotime, while recently identified[30,128] markers of AIRE+ mTECs, such as *NTHL1*, *CALML3*, and *INSM1*, increased either transiently or permanently (Fig. 8i). *ELF3*, *EHF*, and *GRHL1* were expressed broadly within the mTEC II and the secretory mimetic mTEC III clusters independently of *AIRE* expression (Supplementary Fig. 11g). AIRE+ mTEC markers peaked nearly concurrently with those of early

neuronal differentiation, including *ASCL1* and *NEUROG1*, possibly driven by *INSM1*[129] (Fig. 8i). Thereafter, genes abundantly expressed in neurons, such as *STMN2* and *RTN1*, increased especially in mTEC-mim II, and markers more specific to differentiated neuronal subtypes, such as cerebellar Purkinje cells (*PCP4*), as well as glia (*METRN*) and cochlear hair cells (*ATOH1*), appeared in mTEC-mim III (Fig. 8i). Together, these results show that, upon co-culture with thymocytes, our iTECs are capable of recapitulating the dynamics of AIRE+ mTEC and post-AIRE neuronal mimetic mTEC development.

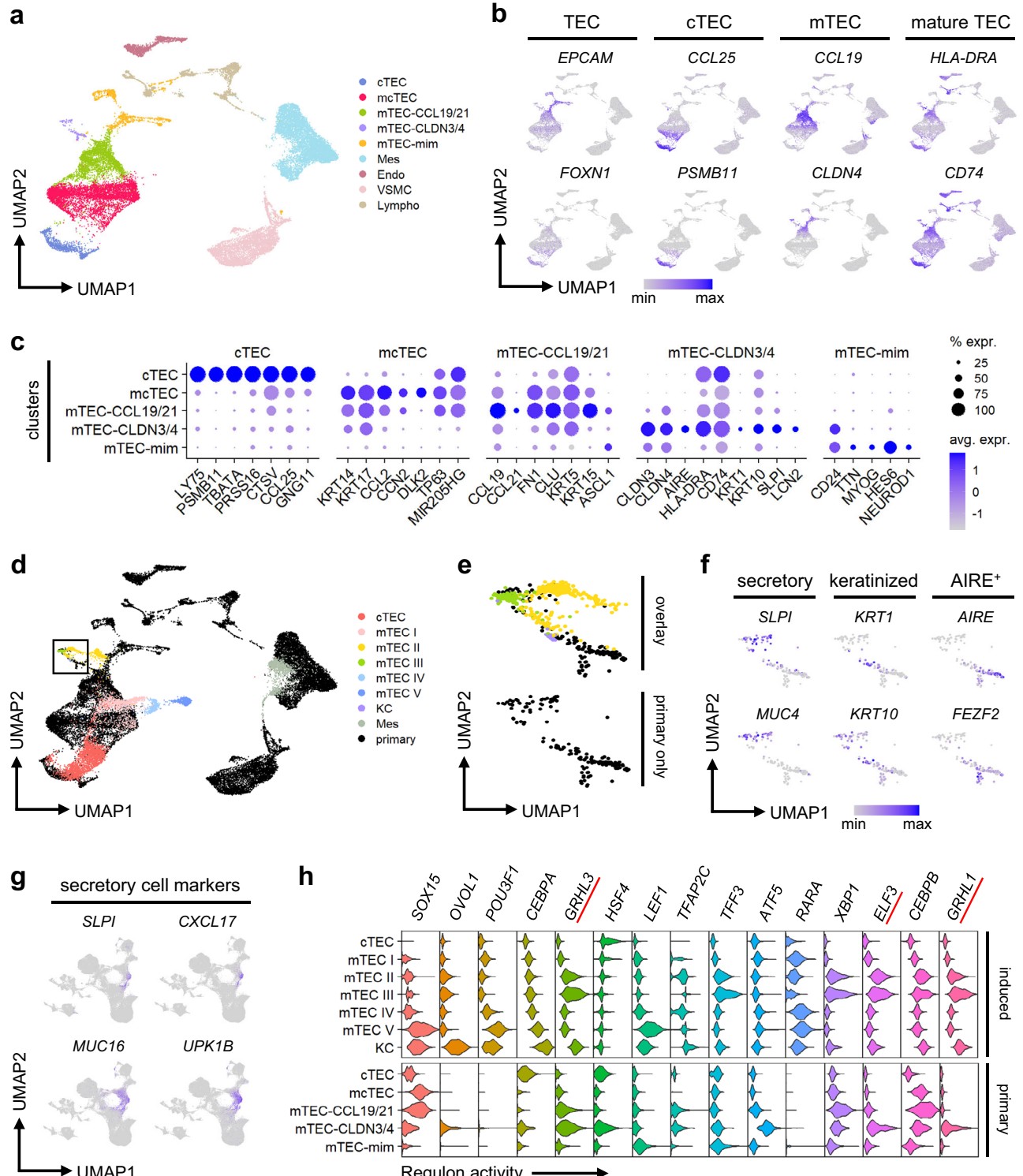

**Fig. 6 | Induced cTEC- and mTEC-like cells resemble human primary TECs.**
**a** UMAP of unsupervised clustering of primary thymic stroma. TEC, thymic epithelial cell; cTEC, cortical TEC; mTEC, medullary TEC; mTEC-mim, mimetic mTEC; mcTEC, intertypical TEC; Mes, mesenchyme; Endo, endothelial cell; VSMC, vascular smooth muscle cell; Lympho, lymphocyte. **b** Expression of TEC markers in primary thymic stroma on UMAP. **c** Dot plot of selected cluster and lineage marker gene expression in the TEC populations of primary thymic stroma. % expr., percentage of cells in which the gene is expressed; avg. expr., average expression. **d**, **e** UMAP of induced TEC, KC, and Mes clusters from day 133, with populations and colors

corresponding to Fig. 5b, integrated with primary thymic stroma (black). The primary mTEC-CLDN3/4 cluster and the co-clustering induced day 133 cells highlighted in (**d**) are shown separately in (**e**). **f** Expression of secretory, keratinized, and AIRE+ mTEC marker genes in the primary mTEC-CLDN3/4 cluster on UMAP.
**g** Expression of secretory cell markers in induced cells on UMAP. **h** Violin plot of regulon activity in each cluster represented by scaled AUC (Area Under the Curve) scores from SCENIC, showing the top 15 induced mTEC marker regulons also found in primary mTECs. All results were obtained using primary thymic stroma from n = 3 pediatric donors and the 201B7 FOXN1mCherry reporter iPSC line.

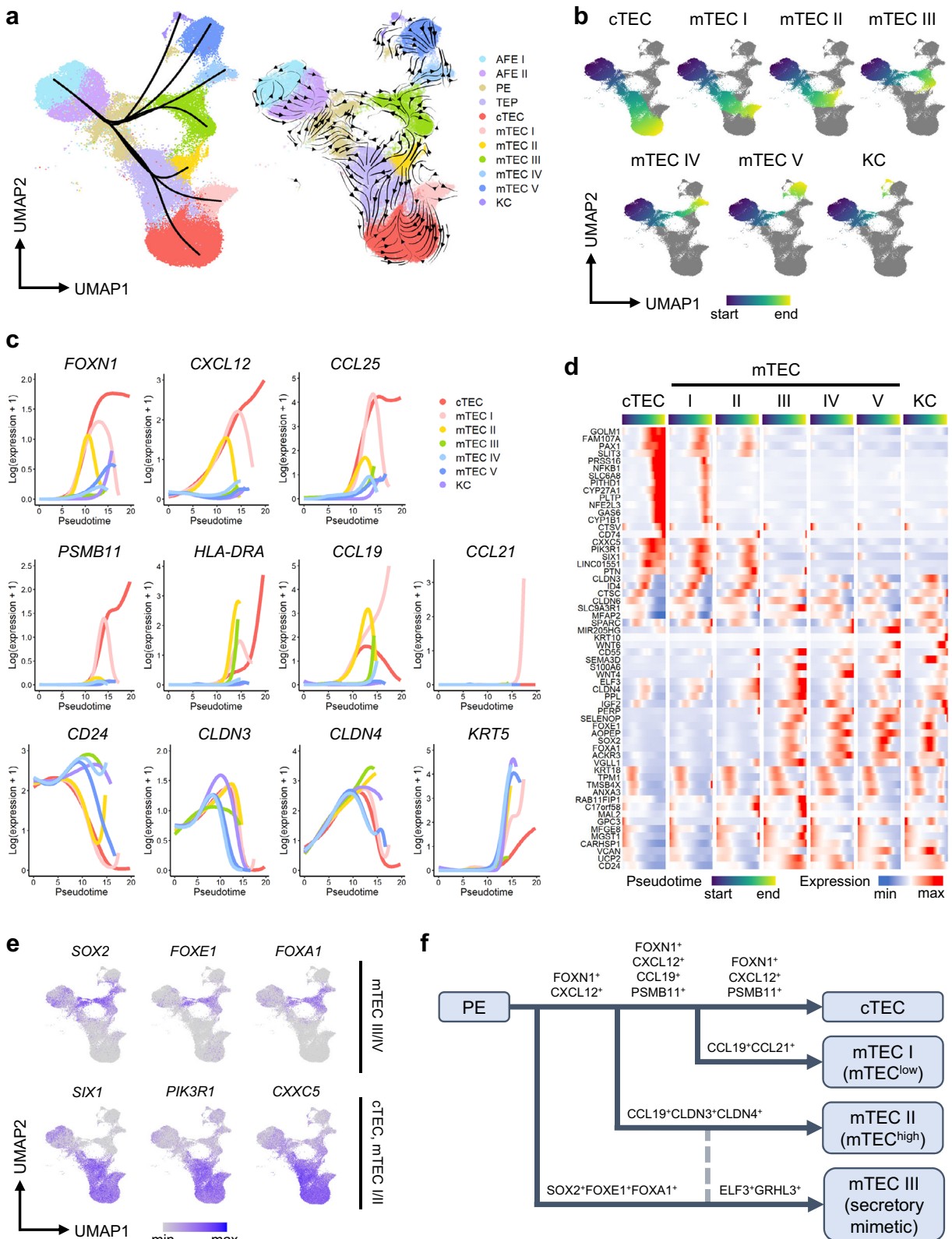

**Fig. 7 | Single cell trajectory analysis of iTECs models TEC development and lineage divergence. a** Slingshot-identified lineage trajectories (left) and RNA velocity by scvelo (right) on UMAP. AFE, anterior foregut endoderm; PE, pharyngeal endoderm; TEP, thymic epithelial progenitor; TEC, thymic epithelial cell; cTEC, cortical TEC; mTEC, medullary TEC; KC, keratinized cell. **b** UMAP showing the trajectories of each lineage from Slingshot colored by pseudotime. **c** Smoothed gene expression of cTEC and mTEC markers along pseudotime in each lineage.

**d** Clustered heatmap of the scaled smoothed gene expression of the top 60 differentially expressed genes along pseudotime using the patternTest in tradeSeq. **e** Expression of selected genes from (**d**) showing the earliest lineage divergence from PE on UMAP. **f** Schematic of TEC lineage divergence inferred from pseudotime analysis of induced TECs (iTEC), displaying the clusters with clear developmental paths and associated lineages in primary TECs. All results were obtained using the 201B7 *FOXN1^mCherry* reporter iPSC line.

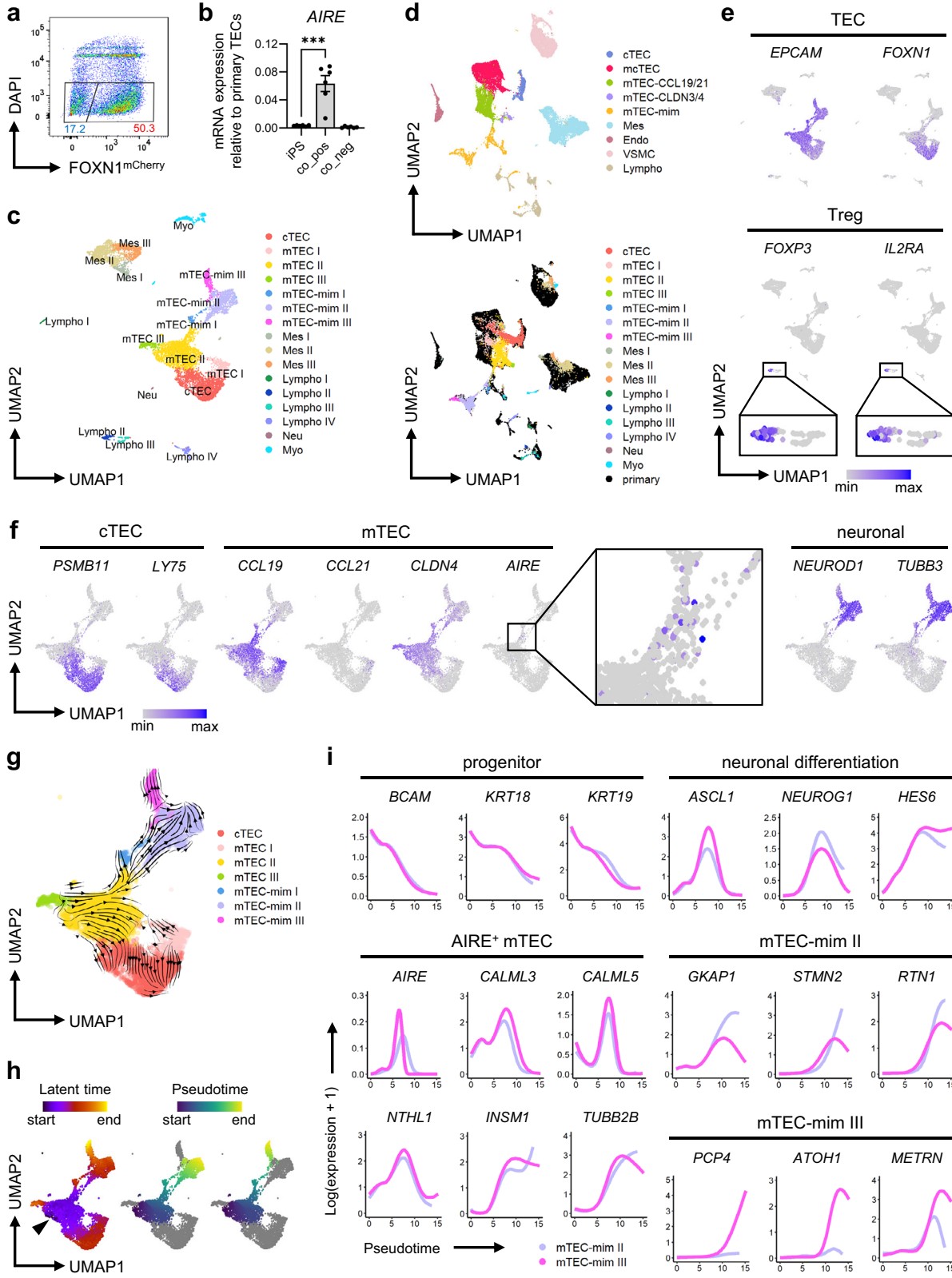

## Discussion

In this study, we successfully established a system to induce diverse, functional TEC-like cells from human iPSCs in vitro. While previous PSC-based protocols have required transplantation, complex 3D co-cultures, or decellularized scaffolds for TEC induction[37–49], our system can produce mature iTECs using simple, chemically defined 2D culture, allowing us to observe their development in a stable, easily

reproducible environment. In an important advance beyond these previous studies, where the induced cells were homogeneous with progenitor-like or intertypical phenotypes, our model recapitulates the divergence of heterogeneous cTEC and mTEC populations from a common progenitor. Through our approach combining RA-based endodermal patterning with a period of self-directed differentiation, we are able to produce highly diverse types of TEC-like cells, which

**Fig. 8 | iTECs recapitulate *AIRE*+ and post-*AIRE* mTEC development in thymocyte co-culture. a** Representative flow cytometry results of iTEC/thymocyte organoids after 2 weeks of co-culture from $n = 6$ independent experiments. TEC, thymic epithelial cell; iTEC, induced TEC. **b** mRNA expression of *AIRE* at the iPSC stage (iPS) and in the mCherry+ (co_pos) and mCherry− (co_neg) populations after 2 weeks of iTEC/thymocyte co-culture. All values indicate the mean ± SEM from $n = 6$ independent experiments. Statistical significance was determined using unpaired two-sided *t*-tests (n.s. no significant difference, *$p < 0.05$, **$p < 0.01$, ***$p < 0.001$). The exact *p*-value is 0.000390. **c** UMAP of unsupervised clustering of cells after iTEC/thymocyte co-culture for 2 weeks from $n = 1$ independent experiment, consisting of 14 pooled organoids from one induction using four thymocyte donors with 3-4 organoids each. cTEC, cortical TEC; mTEC, medullary TEC; mTEC-mim, mimetic mTEC; Mes, mesenchyme; Lympho, lymphocyte; Neu, neuron; Myo, myocyte. **d** UMAP of the co-cultured cells from (**c**) integrated with the primary

thymic stroma from Fig. 6a, showing the clustering of the primary thymic stroma alone (top) and the overlay of the iTEC/thymocyte co-culture cells (bottom). **e, f** Expression of TEC and regulatory T cell (Treg) marker genes in all co-cultured cells (**e**) and cTEC, mTEC, and neuronal marker genes in the *EPCAM*+ clusters (**f**) on UMAP. Smaller areas containing *FOXP3*+*IL2RA*+ cells in (**e**) and the majority of the *AIRE*+ cells in (**f**) are also shown magnified. **g** RNA velocity of the *EPCAM*+ clusters of co-cultured cells by scvelo on UMAP. **h** UMAP showing the latent time inferred from RNA velocity (left) and the pseudotime of the mTEC-mim II and III lineages derived by Slingshot (middle and right). The black arrowhead indicates the area with the earliest latent time. **i** Smoothed gene expression of epithelial progenitor, AIRE+ mTEC, and early neuronal differentiation markers, as well as selected mTEC-mim II and mTEC-mim III cluster markers, along pseudotime in each lineage. All results were obtained using the 201B7 *FOXN1*^mCherry reporter iPSC line co-cultured with primary double-positive (DP) thymocytes.

---

share expression profiles and functional features with their primary counterparts in the pediatric thymus. Although it has long been assumed that signals necessary for TEC differentiation, maturation, and maintenance arise from their 3D microenvironment, as suggested by the rapid downregulation of *FOXN1* in isolated primary TECs placed into 2D culture[34,35], our iTECs stably express *FOXN1* and MHC class II molecules even after 133 days in 2D conditions, indicating that these crucial signals were successfully recapitulated and maintained in our culture. This long-term stability overcomes the limitation of previous induction protocols maintaining TEC-like cells for only up to 30 days with low expression of these markers in 2D when compared to primary TECs[37–49]. Moreover, in an important advance for PSC-based modeling of the thymus, our iTECs are able to differentiate into *AIRE*+ mTEC-like cells and recapitulate *AIRE*-dependent neuronal mimetic mTEC development in thymocyte co-culture, indicating that they are able to respond to signals from thymocytes for further differentiation up to the terminal stages similarly to primary TECs. Together, these capabilities demonstrate the fidelity of our system and provide an opportunity to shed light on developmental processes that have otherwise been exceedingly difficult to observe.

Therefore, we used our system to explore important gaps in our understanding of human thymus organogenesis, especially how the remarkably diverse TEC lineages arise. We found that a precise dose of RA not only specifies PE, as previously reported[130], but also drives the differentiation of the thymic lineage in the absence of posteriorizing signals, as it led to swift upregulation of *FOXN1*, whose expression alone was enough to downregulate *GCM2* over time. This finding is consistent with murine and avian studies suggesting that both an under- and overabundance of RA at the PE stage may disrupt thymus organogenesis[60,131]. Furthermore, since parathyroid-like cells were no longer detected after day 49, this indicates that the thymic lineage may constitute the default lineage in the 3rd PP in the absence of other signals, such as NOTCH inhibition. We were surprised to see a potential mTEC-like lineage divergence as soon as day 18 when *FOXN1* and *GCM2* only started to be expressed, marking the separation of the thymic and parathyroid primordia. In the mouse, separate domains of *Foxn1* and *Gcm2* expression are seen by E11.5[132], when a transient *Foxn1*−*Sox2*+*Foxe1*+ population is also present[12]. We were able to observe cells of this stage over several weeks, likely corresponding to the period up to 6–7 weeks of gestation when both *FOXN1* and *FOXE1* are expressed in the human thymic primordia[4,133]. Our analyses suggest *FOXN1*−*SOX2*+*FOXE1*+ cells may be the earliest progenitors of an *EHF*+*ELF3*+*GRHL1*+ secretory mimetic mTEC-like population, with some similarities to the Sox2+ proximal lung lineage, which also develops under lower BMP signaling and eventually gives rise to secretory and ciliated cells[134,135]. Not all early TFs are shared between these lineages, however, as *FOXE1* is not expressed in the developing lung[133,136,137], and *NKX2-1*, required for lung specification[138], is not present in our system.

After *SOX2*+*FOXE1*+ cells branched off, *FOXN1*+ progenitors by day 28 started assuming characteristics of cTECs, from which the two mTEC-like populations expressing either *CLDN4* or *CCL21* subsequently appeared to diverge. In mice, embryonic progenitors show preferential differentiation to cTECs, while postnatal progenitors assume a bias toward mTECs[27]. Consistent with this, single cell profiling of human thymi has shown that cTECs are more abundant in early fetal samples, while mTECs are the major TEC population in postnatal samples[30]. Taken together with these previous findings, our results indicate that early cTEC-like cells arise before mTEC lineage commitment and may still have multi-lineage potential until the downregulation of *FOXN1* and cTEC markers. In our system, *FOXN1* downregulation coincided with increased expression of mTEC markers in the mTEC I and mTEC II clusters, suggesting the presence of a transcriptional program regulating *FOXN1* during mTEC lineage commitment. Indeed, human primary mTECs have much lower *FOXN1* expression compared to cTECs[35], and *GRHL1*, a potential repressor of *FOXN1*[18], was inferred to be highly active in the mTEC II and mTEC III clusters in our GRN analysis. The mTEC I lineage was the last to diverge from the cTEC lineage in our system, even expressing cTEC-specific *PSMB11* before switching to high expression of mTEC-specific *CCL21* by day 133. This stage likely corresponds to week 11 or 12 in the human fetus, when CCL21+ cells start to arise, but AIRE is not yet expressed[4]. After co-culture with thymocytes, we were able to detect *AIRE*+ mTEC-like cells, possibly corresponding to the earliest AIRE-expressing TECs during week 13 of fetal development[4].

Overall, these data suggest a default pathway to cTECs, from which mTECs sequentially diverge over time during embryonic development, consistent with the asymmetrical serial progression model proposed in mice[3,115,139]. Previous reports have indicated that this serial progression may be exclusive to the embryonic period, given that in the postnatal murine thymus, the mTEC compartment is mostly maintained by unipotent mTEC-restricted stem cells, which arise from *Psmb11*-expressing progenitors only during early development[25,26]. While lineage tracing studies have demonstrated the bipotent potential of embryonic cTEC-like cells[22–24], adult bipotent progenitors have been detected in various subsets including Ly51+MHCII^high Plet1+ and Ly51^low MHCII^low Sca-1^high α6-integrin^high in mice[140,141]. Recently, it has also been shown that embryonic but not postnatal Ccl21+ cells are capable of converting into both cTECs and Aire+ mTECs[142], and embryonic Krt19+ and Sox9+ subsets may act as mTEC-biased progenitors[14,143]. Together, these murine studies indicate multiple possible pathways toward the cTEC and mTEC fates, with differing progenitor identities and levels of plasticity between the fetal and postnatal periods, although whether the same developmental dynamics are present in the human thymus, where certain markers such as *PLET1* are not expressed[144], remains unclear. As iPSC-based systems likely reflect the embryonic rather than the postnatal state, our iTECs offer a unique look into early human thymus

development that postnatal thymic epithelial stem cells or TEC lines cannot provide. Indeed, our single cell profiling allowed us to see developmental shifts in the TEP cluster, with RNA velocity further hinting at plasticity between closely related TEC lineages. While the heterogeneity of the terminally differentiated mimetic population is not yet fully represented in our system, with most mimetic mTEC-like cells by day 133 having a secretory or keratinized phenotype, and all likely post-*AIRE* cells in thymocyte co-culture assuming a neuronal or neuroendocrine phenotype, further improvements in modeling the lymphostromal crosstalk of the thymus in vitro may yet yield other subtypes of mimetic mTECs and allow further exploration of their origins in humans.

Apart from offering important biological insights, our iTECs also show promise for clinical application. For severely immunodeficient patients of DiGeorge syndrome[19], Pignata Guarino syndrome[20], and other disorders of athymia, thymus transplantation is an established life-saving treatment, but it also carries risks of autoimmune complications and mortality through infections before sufficient T cell reconstitution occurs[145,146]. As our system allows the differentiation of mature iTECs capable of generating naïve T cells with diverse TCR repertoires, we can envisage in vitro production of either T cells or thymic epithelium from patient-derived iPSCs or HLA-matched iPSC stocks as an alternative to allogeneic transplantation for immune reconstitution. This approach ensures patient MHC restriction, a major challenge for T cell therapies, and may further be applied to treatments of other genetic immunodeficiencies and cancers[147–149]. The emergence of *AIRE*[+] and *FOXP3*[+] cells after thymocyte co-culture also raises the possibility that iTECs could support negative selection and Treg generation, which will be an important avenue to explore for applications to autoimmune diseases. Additionally, through its capability to recapitulate embryonic TEC development, our system will enable in vitro modeling of congenital athymias to better understand how these pathologies perturb early developmental processes and how they could be alleviated. Thymic dysfunction resulting from congenital diseases, chronic infections, chemotherapy, and age-related thymic involution may be mitigated with drugs promoting TEC proliferation and regeneration[150]. Due to the long-term stability of TEC phenotypes and *FOXN1* expression over several months even in 2D culture, our iTECs will serve as an excellent candidate for screening drugs that could affect thymus development, pathology, and involution.

Taken together, our system, through its capability of generating diverse human cTEC- and mTEC-like cells, which up to now have been highly difficult to obtain from primary or other sources, presents an important platform to unveil the complexities of human thymus biology and is expected to contribute to broad applications in both basic and clinical science.

## Methods
### Collection of human thymus tissue
Thymi were collected from 23 donors, 12 female and 11 male, ranging between 1 month and 4 years of age (median = 4 months), undergoing pediatric cardiac surgery at the Kyoto University Hospital. Collected thymi were placed in cold, sterile PBS for up to 1 h until further processing for stromal cell and thymocyte isolation. This study complies with all relevant ethical guidelines and was approved by the Ethics Committee of Kyoto University Graduate School, Faculty of Medicine, and Kyoto University Hospital. Written informed consent was obtained from the parents of the donors. No compensation was provided. Sex, defined as the sex assigned at birth, was not considered in the study design, and disaggregated sex or gender information was not collected. Both male and female samples were used in each experiment including two or more donors, but due to their low number, the influence of sex on the results cannot be determined.

### iPSC and other cell line culture
The iPSC lines 201B7[63] (human, female, RRID:CVCL_A324, kindly gifted by Dr. Masato Nakagawa, Kyoto University), 409B2[70] (human, female, RRID:CVCL_K092, kindly gifted by Dr. Keisuke Okita, Kyoto University), and 1383D6[71,72] (human, male, RRID:CVCL_UP39, kindly gifted by Dr. Masato Nakagawa, Kyoto University) were previously established with written informed consent and used for this study with the approval by the Ethics Committee of the Center for iPS Cell Research and Application, Kyoto University. The iPSCs were cultured in feeder-free conditions in StemFit AK02N medium (Ajinomoto) at 37 °C and 5% $CO_2$. They were passaged once per week and seeded at a density of 1100 cells/cm$^2$ onto plates coated with iMatrix-511 silk (Nippi). At the time of passage, the medium was supplemented with 10 μM Y-27632 (Nacalai Tesque), which was removed on the following day. The previously established stromal cell line MS5-hDLL1[92] (mouse, male, RRID:CVCL_VR88, purchased from Sigma-Aldrich, Cat#SCC167) was passaged twice a week at a ratio of 1:6 and maintained in D-MEM (Wako) supplemented with 10% fetal bovine serum (FBS, Sigma-Aldrich) and 1% Penicillin-Streptomycin Mixed Solution (Nacalai Tesque) at 37 °C and 5% $CO_2$. 201B7, 409B2, and 1383D6 were authenticated by STR analysis at the time of deposition to the RIKEN BioResource Research Center, and their *FOXN1*[mCherry] reporters were authenticated by karyotyping by Chromocenter (Yonago, Japan) within five passages of their establishment. MS5-hDLL1 was verified by the vendor to be of mouse origin by a Contamination Clear panel by Charles River Animal Diagnostic Services.

### Isolation of human primary thymic stroma
Similarly to a previously published protocol[151], thymi were manually cut into 2–3 mm pieces and enzymatically digested for 60–90 min in Liberase solution containing 100 μg/ml Liberase TM (Roche), 200 μg/ml DNase I (Worthington), and 1% FBS in RPMI 1640 (Wako) at 37 °C on the Eppendorf ThermoMixer C with 800 rpm mixing frequency. Upon complete dissolution, the cells were centrifuged at 350 x g for 5 min, and the supernatant was removed. The cell pellet was resuspended in 83% v/v Percoll (Cytiva) in PBS, after which a lighter layer of 47% v/v Percoll in PBS and a further layer of only PBS were overlaid. The resuspended cells were centrifuged at 1490 x g for 30 min, 4 °C, and the stromal cells enriched between the PBS and lighter Percoll layer were collected. Cells were resuspended in TEC buffer containing 1% FBS and 5 mM EDTA (Dojindo) in PBS, counted, and incubated on ice with 50 μl MojoSort Human CD45 Nanobeads (BioLegend) per 1 ml TEC buffer for 1 × 10$^8$ cells for 15 min. After one wash in TEC buffer, cells were resuspended in TEC buffer and depleted of CD45$^+$ cells on a MagniSort Magnet (eBioscience) for 5 min. The stromal CD45$^-$ fraction was collected and frozen for subsequent purification by flow cytometry.

### TEC induction from iPSCs
iPSCs were seeded at a density of 1300–3200 cells/cm$^2$ onto plates coated with iMatrix-511 silk in AK02N medium supplemented with 10 μM Y-27632. On the next day, the medium was changed to AK02N only. Four days after seeding, the induction was started using the same medium and factors as previously reported for AFE induction[50] up until day 7.

Briefly, the base medium was changed to CDM2, which is composed of a 1:1 mixture of IMDM (Thermo Fisher) and F-12 (Thermo Fisher), with added 1 mg/ml polyvinyl alcohol (Sigma-Aldrich), 0.7 μg/ml insulin (Wako), 15 μg/ml transferrin (Nacalai Tesque), 450 μM monothioglycerol (Sigma-Aldrich), and 1:100 chemically defined lipid (Thermo Fisher). On day 0, CDM2 was supplemented with 100 ng/ml Activin A (Nacalai Tesque), 2 μM CHIR99021 (Nacalai Tesque), and 50 nM PI-103 (Tocris) to induce primitive streak (PS) cells. Activin A is known to drive pluripotent stem cells toward the PS and its derivatives when PI3K signaling is suppressed[152] and WNT signaling is active[50].

PI-103, a potent PI3K inhibitor, and CHIR99021, a GSK3 inhibitor promoting WNT activation, are capable of effectively specifying the PS[50]. On day 1, the medium was changed to CDM2 with 100 ng/ml Activin A and 250 nM LDN-193189 (Reprocell) to induce DE. Since BMP signaling at this stage leads to the formation of mesoderm[153], BMP inhibitor LDN-193189 is used to specifically induce DE[50]. On day 3, the medium was changed to CDM2 with 250 nM LDN-193189 and 1 μM A-83-01 (Tocris) for AFE induction. The AFE fate is promoted by the combined inhibition of BMP and TGFβ signaling[154], with A-83-01 being a potent inhibitor of the TGFβ pathway effective at specifying AFE[50].

Next, to induce PE from AFE, the medium was changed to CDM2 supplemented with 150 nM RA (Sigma-Aldrich) and 50 ng/ml FGF8 (R&D Systems) on day 7. RA is responsible for the patterning of the posterior pharyngeal pouches[58–60] and FGF8 promotes the normal development and morphology of the pharyngeal arches[61,62]. From day 18 onwards, medium changes were done using CDM2 only to allow for undirected differentiation of all 3rd PP derivatives. The medium was changed every 2–3 days. Throughout the induction, cells were kept in the same plates without passage. Details of all reagents are in Supplementary Data 1.

### Establishment of *FOXN1^mCherry^* reporter lines

*FOXN1^mCherry^* reporters were established from the iPSC lines 201B7, 409B2, and 1383D6 using the CRISPR-Cas9 system. Guide RNA (gRNA) with spacer sequence 5′-GCTCCAGCGTTTGCCTGGTC-3′ was designed to target the 3′ UTR near the *FOXN1* stop codon. DNA templates for the gRNA were amplified with PCR using Phusion High-Fidelity DNA Polymerase (Thermo Fisher) and purified using the FastGene Gel/PCR Extraction Kit (Nippon Genetics). gRNA was transcribed in vitro using the MEGAshortscript Kit (Thermo Fisher) and purified using the RNeasy MinElute Cleanup Kit (QIAGEN). The donor plasmid DNA, containing the homology arms, the T2A-mCherry-pA sequence, and a puromycin resistance cassette under the EF1α promoter flanked by two loxP sites, was assembled using the In-Fusion HD Cloning Kit (Takara Bio) with the pENTR-Donor-MCS2 vector[155] as the backbone. The Competent High DH5α Kit (Toyobo) was used for transformation of *E. coli* with the vector, which was purified using the NucleoBond Xtra Midi EF Kit (Takara Bio).

For genome editing, using previously optimized conditions[156], 1 μg donor plasmid DNA, 1.25 μg gRNA, and 5 μg TrueCut Cas9 Protein v2 (Thermo Fisher) were transfected into the iPSC lines with the P4 Primary Cell 4D-Nucleofector X Kit S (Lonza). After 3–5 days, 0.5 μg/ml puromycin (Nacalai Tesque) was added to select for candidate clones incorporating the insert. Emerged colonies were randomly picked, expanded, and treated with Cre Recombinase Gesicles (Takara Bio) to remove the floxed puromycin cassette. The resulting cells were re-seeded sparsely, colonies were randomly picked again, and each resulting clone was analyzed by PCR with PrimeSTAR GXL DNA Polymerase (Takara Bio) after DNA extraction with the NucleoSpin Tissue XS Kit (Macherey-Nagel) to determine the presence of the insert. After gel electrophoresis, bands were detected using the FAS-Digi (Nippon Genetics) with an attached ILCE-5100 camera (SONY). Sequencing was performed by Eurofins Genomics (Tokyo, Japan) to confirm the absence of indels in the allele without the insert and the presence of T2A-mCherry in the allele with the insert. Primers used for PCR and sequencing are listed in Supplementary Data 2. Karyotyping was performed by Chromocenter (Yonago, Japan) to confirm the cell identity and the absence of major chromosomal alterations. All *FOXN1^mCherry^* reporter iPSC lines generated in this study are available upon request.

### Flow cytometry

The frozen CD45⁻ thymic stromal cells were defrosted and washed once in TEC buffer. Cells were stained with antibodies diluted 1:50 in TEC buffer for 15 min on ice. After washing with TEC buffer, the cells were transferred to 5 ml tubes through a 35 μm mesh for analysis with

the BD FACSAria II. Propidium iodide (PI, Nacalai Tesque) was used at 1 μg/ml to stain dead cells. To collect primary TECs for qPCR controls, the EPCAM⁺CD31⁻CD45⁻CD235ab⁻ fraction was sorted, centrifuged for 10 min at 150 x g, 4 °C, and resuspended in TRIzol Reagent (Thermo Fisher). To collect primary thymic stroma for scRNA-seq, the EPCAM⁻CD45⁻CD235ab⁻ and EPCAM⁺CD31⁻CD45⁻CD235ab⁻ fractions were sorted separately, centrifuged, resuspended in 0.04% UltraPure BSA (Thermo Fisher) in PBS, and mixed at a ratio of 1:2 to obtain stromal cells enriched for TECs.

For flow cytometry of induced cells on days 3 and 7, the cells were first washed with PBS and dissociated using Accutase (Nacalai Tesque) for 7–10 min at 37 °C. Cells were detached with cell scrapers and washed twice with FACS buffer consisting of 2% FBS in PBS. For cells on day 3, antibodies diluted 1:50 in FACS buffer were used to stain for 15 min on ice. After two washes in FACS buffer, cells were passed through a 35 μm mesh for analysis with the Cytek NL-3000. DAPI was used at 0.5 μg/ml to stain dead cells. For cells on day 7, Ghost Dye Red 710 (Tonbo Biosciences) was used at 1:1000 in PBS to stain dead cells for 30 min on ice before proceeding to further staining. After one wash in FACS buffer, cells were fixed with Fixation/Permeabilization solution (Thermo Fisher) at room temperature for 30 min. Cells were washed twice using Permeabilization Buffer (Thermo Fisher) and stained with antibodies diluted 1:10 (FOXA2) or 1:100 (SOX2) in Permeabilization Buffer for 30–45 min at room temperature. After two washes with Permeabilization Buffer, cells were resuspended in FACS buffer, passed through a 35 μm mesh, and analyzed with the Cytek NL-3000.

For flow cytometry of induced cells on day 28–133, the cells were washed with PBS and dissociated using Accumax (Nacalai Tesque) for 20–30 min at 37 °C. Cells were detached with cell scrapers and collected in 50 ml tubes containing FACS buffer consisting of 0.1% bovine serum albumin (BSA, Nacalai Tesque) in PBS. Large cell clumps that could not be dissociated even after pipetting were allowed to sink to the bottom of the tubes for 30–60 s and the supernatant was transferred to new tubes. Cells were centrifuged for 5 min at 350 x g, 4 °C, washed once with FACS buffer, and passed through a 35 μm mesh. Antibodies diluted 1:10 in FACS buffer were used for staining on ice for 20 min. After washing once and resuspending the cells in FACS buffer, analysis was performed using the BD FACSAria II. DAPI was used at 0.5 μg/ml to stain dead cells. To collect induced cells for qPCR, the mCherry⁻, mCherry^low, and mCherry^high fractions were sorted separately, centrifuged for 10 min at 150 x g, 4 °C, and resuspended in TRIzol Reagent. For scRNA-seq, the same fractions were sorted, centrifuged, resuspended in 0.04% UltraPure BSA in PBS, and mixed at a ratio of 1:1:2 to enrich for mCherry⁺ cells.

For flow cytometry of induced cells on days 7, 11, and 18 for scRNA-seq, the cells were processed and analyzed in the same way as described above for cells on day 28–133 with varying lengths of Accutase or Accumax dissociation (7–20 min) depending on cell density, but due to the absence of mCherry⁺ cells, the entire live fraction was sorted and resuspended in 0.04% UltraPure BSA in PBS.

Flow cytometry results were analyzed in BD FACSDiva Software, SpectroFlo, FlowJo, and Microsoft Excel. Gating strategies are shown in Supplementary Fig. 12 and 13. All antibodies, together with their dilutions, catalog numbers, and further details, are listed in Supplementary Data 3.

### Organoid culture and TCR repertoire analysis

Human primary thymocytes were isolated by gently vortexing 2–3 mm thymus pieces, after which cells from the supernatant were incubated in RBC Lysis Buffer (BioLegend) for 10 min on ice, washed in 1% FBS in RPMI 1640, and frozen for future use. Stocked thymocytes were defrosted, washed in 1% FBS in PBS, and stained with antibodies diluted 1:10 in the same buffer for 20 min on ice. After washing once more in 1% FBS in PBS, cells were passed through a 35 μm mesh and analyzed using the BD FACSAria II or BD FACSymphony S6. PI was used at 1 μg/ml to

stain dead cells. To isolate DP and DN thymocytes for organoid co-culture, the CD4+CD8α+CD45+CD69- and CD5+CD7+CD4-CD8α-CD11c-CD14-CD19- populations were respectively sorted and resuspended in RPMI 1640 with 4% B-27 Supplement (Thermo Fisher). To isolate SP thymocytes as donor controls for TCR repertoire analysis ($n = 1$ pediatric donor, male), the CD45+CD3+TCRαβ+CD4+CD8α- and CD45+CD3+TCRαβ+CD4-CD8α+CD8β+ populations were sorted at 10,000 cells each and resuspended in ISOGEN-LS (Nippon Gene).

For organoid co-culture, 10,000 DP or 5,000 DN thymocytes and 100,000 MS5-hDLL1 cells or 100,000 sorted mCherry+ or mCherry- iTECs from day 37 ± 2 of induction were mixed, resuspended in 2.5 μl Matrigel (Corning) per organoid, and seeded onto 0.8 μm membranes (Cytiva) floating in 12 well plates filled with RPMI 1640, supplemented with 10 μM Y-27632, 5 ng/ml FLT3L (PeproTech), 5 ng/ml IL7 (Pepro-Tech), 30 μM L-ascorbic acid 2-phosphate (Sigma-Aldrich), and 4% B-27 Supplement. Y-27632 was removed on the following day, after which the medium was changed every 2–3 days. To collect thymocytes for analysis after co-culture, organoids were homogenized using pestles. Cells from the supernatant were washed in 1% FBS in PBS and stained with anti-bodies diluted 1:10 in the same buffer for 20 min on ice. After one more wash, cells were passed through a 35 μm mesh and analyzed with the Cytek Aurora. For TCR repertoire analysis of the SP thymocytes obtained after mCherry+ iTEC co-culture with DP thymocytes, the CD45+CD3+TCRαβ+CD4+CD8α- and CD45+CD3+TCRαβ+CD4-CD8α+CD8β+ populations were sorted at 10,000 cells each using the BD FACSAria II and resuspended in ISOGEN-LS.

TCR repertoire analysis was performed by Repertoire Genesis (Osaka, Japan) as previously described[157,158]. Briefly, RNA was extracted using the RNeasy Lipid Tissue Mini Kit (Qiagen) and reverse tran-scribed with Superscript III reverse transcriptase (Thermo Fisher). TCRβ genes were amplified with adapter-ligation PCR[157,158], index sequences were added using the Nextera XT Index Kit v2 Set A or Set D (Illumina), and sequencing was performed with the MiSeq (Illumina). Sequences of TCRβ genes *TRBV* and *TRBJ* were assigned based on the international ImMunoGeneTics information system (IMGT) database[159]. All data processing was performed using Repertoire Gen-esis analysis software[157,158]. *TRBV* and *TRBJ* gene usage percent was calculated as the frequency of reads in total sequence reads. For the estimation of TCR repertoire diversity, the Shannon index[97] was cal-culated with the Repertoire Genesis analysis software.

For qPCR and scRNA-seq, organoids of mCherry+ iTECs and DP thymocytes co-cultured for 14 days were dissociated in Accumax for 15 min at 37 °C, washed in 1% FBS in PBS, and analyzed using the BD FACSymphony S6. DAPI was used at 0.5 μg/ml to stain dead cells. For qPCR, the mCherry- and mCherry+ populations were sorted, cen-trifuged, and resuspended in TRIzol Reagent. For scRNA-seq, the mCherry- and mCherry+ populations were sorted, centrifuged, resus-pended in 0.04% UltraPure BSA in PBS, and mixed at a ratio of 1:2. This scRNA-seq dataset includes thymocytes from $n = 4$ pediatric donors, 3 female and 1 male.

## RNA extraction and quantitative PCR
RNA was extracted from cells resuspended in TRIzol Reagent using the Direct-zol RNA MicroPrep Kit (Zymo Research). 300 ng of extracted total RNA was reverse transcribed using the PrimeScript RT Reagent Kit with gDNA Eraser (Takara Bio). The cDNA was diluted 10 times and used in duplicate or triplicate reactions for qPCR with the THUNDER-BIRD Next SYBR qPCR Mix (Toyobo). The results were analyzed with the QuantStudio 12K Flex Real-Time PCR System (Thermo Fisher), Microsoft Excel, and GraphPad Prism. All results were normalized by *ACTB* expression. For all mRNA expression values of induced cells shown as relative to primary TECs, the values were normalized by the mean mRNA expression values of FACS-purified primary TECs (EPCAM+CD31-CD45-CD235ab-) from $n = 4$ pediatric donors, 2 male and 2 female.

## Immunofluorescence staining and microscopy
Induced cells in 24 well plates were washed once in PBS and fixed in 4% paraformaldehyde solution (PFA, Nacalai Tesque) for 15 min at room temperature. After two PBS washes, cells were permeabilized for 15 min at room temperature using 0.4% Triton X-100 (Sigma-Aldrich) in PBS. Cells were washed once in PBST consisting of PBS with 0.1% Tween 20 (Nacalai Tesque) and blocked for 1 h at room temperature with blocking buffer consisting of 1% BSA in PBS. Staining was per-formed overnight at 4 °C with primary antibodies diluted in blocking buffer. For day 18, EPCAM, TBX1, and HOXA3 antibodies were all used in 1:300 dilutions. For days 28, 80, and 133, the dilutions were 1:150 for CLDN4 and KRT5 (Proteintech), 1:600 for KRT8, PSMB11, EPCAM, HLA-DR, KRT5 (BioLegend), VIM, and Ki-67, and 1:1500 for mCherry. On the next day, cells were washed three times in PBST and stained with secondary antibodies diluted 1:500 and DAPI at 0.2 μg/ml in blocking buffer for 1 h at room temperature. After one PBST and two PBS washes, cells were stored in PBS at 4 °C until observation using the Keyence BZ-X810.

All phase-contrast images of non-stained and live cells were also taken using the same microscope. For the structured illumination microscopy of 3D-like nodules, images with a pitch of 1.2 μm over a range of 50–150 μm were taken using the sectioning and Z-stack functions of the Keyence BZ-X810. 3D displays of Z-stacks, repre-sentative of $n = 4$ (live cells) or $n = 3$ (fixed, stained cells) independent experiments with 3 or more imaged nodules each, were created using the Keyence BZ-X810 analysis software, recorded using OBS Studio, and arranged into videos using Microsoft Clipchamp. For the 3D nodules with co-staining of VIM and mCherry, the original red color of the mCherry staining was changed to magenta for viewer accessibility.

## Cell and library preparation for scRNA-seq
Sorted induced cells and primary thymic stroma were applied at 10,000 cells per sample to the Chromium Next GEM Chip K (10x Genomics). Reverse transcription, cDNA amplification, and library construction were performed using the Chromium Next GEM Single Cell 5' GEM Kit v2, Chromium Next GEM Single Cell 5' Gel Bead Kit v2, Library Construction Kit, and Dual Index Kit TT Set A according to the manufacturer's instructions (10x Genomics). Sequencing was per-formed by the CiRA Foundation (Kyoto, Japan) using the NovaSeq 6000 (Illumina) at 50,000 read pairs per cell.

## scRNA-seq data processing and integration
FASTQ files were made and aligned with Cell Ranger using default settings and the refdata-gex-GRCh38-2020-A human reference pro-vided by 10x Genomics. Doublets identified by Scrublet[160] were excluded from further analysis, which was performed with the R package Seurat[161]. Cells with total RNA counts and unique feature counts in the top or bottom 5%, as well as cells containing > 10% mitochondrial counts, were filtered out. This resulted in datasets ranging from 5782–9207 cells per sample and 747–8357 unique fea-tures per cell. SCTransform was performed to normalize and scale the data, while regressing out the mitochondrial mapping percentage and cell cycle score.

All samples of the main induction from day 7–133 were merged and PCA was run on the merged object using the union of the top 3000 variable features of each sample as input. The top 50 PCs were retained for UMAP dimensional reduction and clustering with the Louvain method (resolution = 0.5). Clusters were manually annotated based on the expression of canonical markers of each cell type and the module scores of cTEC and mTEC markers defined by Park et al.[30]. When the same cell type was assigned to more than one cluster, clusters were arbitrarily numbered using Roman numerals I to VI. For primary thymic stroma, obtained from $n = 2$ female and $n = 1$ male pediatric donors, samples were integrated with anchors identified using RPCA. PCA was run on the integrated object using the top 2000 variable features, with

the top 100 PCs kept for UMAP dimensional reduction and clustering with the Louvain method (resolution = 0.1). Several small lymphocyte clusters containing T cells, B cells, and others were merged into one Lympho cluster for visual clarity. For the iTECs co-cultured with DP thymocytes in organoids, PCA was run using the top 2000 variable features, with the top 100 PCs being retained for UMAP dimensional reduction and clustering with the Louvain method (resolution = 0.4). Clusters were manually annotated based on the expression of known markers and the co-clustering with the day 133 induced cell and primary thymic stroma datasets above, as well as the dataset of thymic epithelium reported in Park et al.[30].

To compare induced monoculture or co-culture and primary TECs, SCTransformed samples of induced cells from day 133, primary thymic stroma, and organoid co-cultured cells were each integrated with RPCA using the top 2000 variable features, after which their original cluster annotations were visualized on UMAP. To compare cells from day 38 and 81 of two different inductions, the two SCTransformed samples of the respective induction day were merged, PCA was run using the top 50 PCs with the union of the top 3000 variable features as input, and the result was visualized using UMAP. For subclustering of the TEP cluster of the merged day 7–133 induced dataset, as well as the mTEC II cluster of the iTEC with DP thymocyte co-culture dataset, PCA and UMAP were re-run, clustering was performed with resolution = 0.3 and 0.2, respectively, and the new cluster annotations were added back onto the original UMAP. To compare our co-cultured iTECs with the dataset of thymic epithelium by Park et al.[30], the published raw count matrix and cell annotations were downloaded from Zenodo (https://doi.org/10.5281/zenodo.5500511). The Park et al. dataset was subset for all cells annotated as thymic epithelium and SCTransformed identically to our dataset. Both datasets were integrated with CCA using the top 2000 variable features and visualized on UMAP with each of their original cluster annotations.

### Differential expression and GRN analysis
Cluster markers were found using the Seurat function FindAllMarkers with the Wilcoxon Rank Sum test. All genes with Bonferroni-adjusted $p$-value < 0.05 and average $\log_2$ fold change > 0.25 compared to other clusters were deemed to be markers of that cluster. To assess the similarity of gene expression patterns in induced TECs and the Park et al. dataset[30], the reported top 20 markers of the cTEC cluster and the combined top 20 markers of the mTEC clusters were imported into a module and a score of each module in induced cells was calculated with Seurat's AddModuleScore function. Module scores were visualized on UMAP.

For GRN analysis, all cells of the TEC lineages were subset and the SCENIC pipeline[162] was run with default parameters. GRNs are computational models of TFs regulating each other and their target genes, representing their interactions as networks. GRNs thereby allow the inference of transcriptional states and cell identities. GRNBoost2[163] was used to calculate putative GRNs based on gene co-expression, after which only direct-binding interactions were retained based on known TF-binding motifs. The activity of the regulons, which include the TFs and their potential target genes, was then scored using AUCell. The AUC (Area Under the Curve) scores for each cell were applied to the Seurat objects, and cluster-specific regulons were determined using the FindAllMarkers function. Regulons with Bonferroni-adjusted $p$-value < 0.05 and average $\log_2$ fold change > 0.01 were deemed to be markers of that cluster.

### Trajectory and RNA velocity analysis
Trajectory inference and pseudotime analysis were performed in R with Slingshot[164]. For the induced cells from day 7–133, the root was set as the AFE I cluster and end points were set as the cTEC, mTEC I, mTEC II, mTEC III, mTEC IV, mTEC V, and KC clusters, due to their respective presence on days 7 and 133. To determine genes differentially

expressed along pseudotime between different lineages, the fitGAM and patternTest functions in tradeSeq[165] were used. Genes with wald-Stat > 100 and FDR-adjusted $p$-value < 0.05 were deemed to be differentially expressed. Differentially expressed genes were visualized using the plotSmoothers function in tradeSeq and the Heatmap function in ComplexHeatmap[166]. For RNA velocity analysis, velocyto[167] was used to recount spliced and unspliced reads by considering the mapping strands based on the BAM files generated by Cell Ranger. These count matrices were then applied to the preprocessed Seurat objects. The induced TEC lineage clusters from day 7–133 with the top 2000 variable genes were subset and used as input to the scvelo pipeline[127], which was run with default parameters using the dynamical model. The resulting velocity vector field was displayed on UMAP with scvelo.

For RNA velocity analysis of the iTEC with DP thymocyte co-culture dataset, the *EPCAM*⁺ iTEC cluster with the top 1000 variable genes was subset and analyzed with the scvelo pipeline as described above. The latent time inferred by scvelo was used to set the root as the mTEC II early subcluster and the end points as the mTEC-mim II and mTEC-mim III clusters for Slingshot pseudotime analysis of the neuronal mimetic mTEC-like cells. The fitGAM and associationTest functions in tradeSeq were used to determine the genes whose expression is associated with pseudotime, with mean log fold change > 0.5, waldStat > 10, and FDR-adjusted $p$-value < 0.05 being used as criteria. Genes were visualized using the plotSmoothers function in tradeSeq.

### Ligand-receptor interaction analysis
The induced cTEC, mTEC I, mTEC II, mTEC III, mTEC IV, mTEC V, KC, and Mes clusters from the day 133 sample were subset and used as input for CellPhoneDB[168] with the statistical analysis method. Predicted ligand-receptor interaction pairs were ranked by their interaction score. Dot plots of interaction means and $p$-values were created with a manually curated selection from the top 50 interactions of ligands from the induced Mes cluster and receptors from the induced TEC lineage clusters.

### Statistical analysis and reproducibility
All statistical analyses of scRNA-seq data were conducted within R, Python, and their respective packages as described in the Methods section. No custom algorithms or software were used. For statistical analyses of mRNA and protein expression data, unpaired two-sided $t$-tests were performed in Microsoft Excel and GraphPad Prism. Results with $p$-value < 0.05 were considered significant. Definitions of the center and dispersion, as well as other details, are given in the figure legends. "n" is defined as the number of independent experiments (biological replicates)—i.e., the number of separate inductions in the case of iPSCs and the number of donors in the case of primary cells—and its value for each experiment is provided in the figure legends. No data randomization, blinding, sample size estimation, test of statistical assumptions, or data exclusion was done.

### Reporting summary
Further information on research design is available in the Nature Portfolio Reporting Summary linked to this article.

## Data availability
scRNA-seq data of induced and primary TECs were deposited into the GEO database under the accession number GSE275981 and are publicly available at the following URL: https://www.ncbi.nlm.nih.gov/geo/query/acc.cgi?acc=GSE275981. RNA-seq data of TCR repertoire analysis were deposited into the GEO database under the accession number GSE275815 and are publicly available at the following URL: https://www.ncbi.nlm.nih.gov/geo/query/acc.cgi?acc=GSE275815. Source data are provided with this paper.

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

## Acknowledgements

We are grateful to Erina Yamaguchi, Yurika Matsuda, Tomoko Ikari, and Miho Shimada for experimental assistance, to Satoko Sakurai for assistance with scRNA-seq data processing, to Tomoya Matsumoto for imaging assistance, to Dr. Izumi Ohigashi and Dr. Yousuke Takahama for technical help, and to Dr. Yoshiya Kawaguchi for helpful discussions. This study was supported by JSPS KAKENHI Grant Numbers JP23KK0136 to Y.H., T.Y., and Y.P., JP22J13683 to Y.P., JP22KJ1841 to Y.P., JP25K18473 to Y.P., and JP20K06469 to K.Kometani; AMED Grant Numbers JP23bm1323001 to Y.H., T.Y., and A.H., JP20bm0704043 to Y.H., and JP23bm1123022 to Y.H.; JST SPRING Grant Number JPMJSP2110 to Y.G. and K.Kanai; the JSPS WISE program "The Graduate Program for Medical Innovation (MIP)" to Y.G.; the Takeda Science Foundation to Y.H.; and the iPS Cell Research Fund to Y.H.

## Author contributions

Conceptualization, Y.P. and Y.H.; Methodology, Y.P., K.Kometani, H.X., and A.H.; Software, Y.P. and T.Y.; Validation, Y.P., K.Kanai, M.O., and K.N.; Formal Analysis, Y.P., Y.G., and T.Y.; Investigation, Y.P., Y.G., and K.Kanai; Resources, T.Y., T.I., and A.H.; Data Curation, Y.P.; Writing – Original Draft, Y.P.; Writing – Review & Editing, Y.P. and Y.H.; Visualization, Y.P.;

Supervision, Y.H.; Project Administration, Y.P. and Y.H.; Funding Acquisition, Y.P., K.Kometani, and Y.H.

## Competing interests

Y.P., Y.G., and Y.H. are named inventors of patent application no. PCT/JP2023/021888, related to the induction method of thymic epithelial cells, filed by Kyoto University. The remaining authors declare no competing interests.
