## [Transparent Peer Review file · Nature Communications]

An iPSC-based in vitro model recapitulates human thymic epithelial development and multi-lineage specification

Corresponding Author: Professor Yoko Hamazaki

Version 0:

Reviewer comments:

Reviewer #1

(Remarks to the Author)

The authors established a human iPSC-based model to study thymus organogenesis using a simple 2D culture system. The model demonstrates that controlled RA treatment can guide PE to adopt a thymic fate, generating FOXN1+ cells.

Additionally, the system produces diverse TEC-like populations through self-directed differentiation over a period of 130 days. The induced TEC-like cells also support in vitro generation of naïve T cells. Single-cell gene expression profiles further illustrate that this model generates diverse TEC-like populations, including cortical and medullary subtypes comparable to primary human TECs.

The experimental design is well-constructed, and the conclusions drawn from the data appear appropriate. However, several major and minor issues remain that should be addressed to strengthen the manuscript's claims.

<1> In figure 1, a key finding may be the induction of PE through the addition of FGF8 and retinoid acid, allowing cells progress from the PE stage through self-directed differentiation toward TEP and subsequently into TECs by culturing over 130 days.

While this strategy is effective, it is worth noting that similar concentrations of RA and FGF8 have been previously employed in studies using ES cells to drive differentiation to the PE stage, which may slightly reduce the novelty of this approach. However, a notable strength of this study is the extended period of over 130 days for self-directed differentiation beyond the PE stage, allowing the maturation of diverse TEC-like populations in vitro.

1) Based on this, an explanation (using data) of how self-directed differentiation can be achieved in vitro might be more important than emphasizing controlled retinoic acid treatment.

2) At later time point, diverse TEC populations were observed, including cTECs and mTECs. The differentiation of these populations may require various signaling pathways, such as NOTCH signaling, LTBR signaling, and NF-κB signaling. During the differentiation process (PE - TEP - TEC), were these signalling-related molecules observed in a time dependent manner?

3) Figure 1i is not clear to understand. Proper legend in the figure should be added.

<2> In the thymus, the cortical and medullary regions are distinct, with specific TEC subpopulations observed in each region. In this model, are both cortical and medullary regions observed? In figure 2 and supplementary figure 3/4, each marker is visualised together with FoxN1, only. Co-staining with general cTEC and mTEC markers simultaneously is required to determine whether this model also recapitulate the spatial distribution of cTECs and mTECs.

It is mentioned that "By day 133, 3D-like nodules up to 100 μm in height were observed within mCherry+ colonies.". What do these high 3D-like nodules represent? How are the expression of cTEC or mTEC markers associated with the formation of these 3D-like nodules?

<3> It could be more impactful if gene editing were used to generate a lineage tracing system for FOXN1 (for example, using FOXN1 lineage with GFP and FOXN1 expression with mCherry). This would further support the author's claim regarding the mimetic mTECs (from lines 383 to 387), suggesting that some mTEC differentiation occurs independently of FOXN1 expression.

<4> After iPSCs are induced into TEPs, are there any proliferating subpopulations during further differentiation process? Can TEPs regenerate or self-renew, as previously described?

<5> The absence of AIRE expression at all time points raises a significant concern and weakness in the study, as functional TEC differentiation should inherently include AIRE-expressing mTECs.

1) In the scRNA-sequence dataset, the authors used the reference dataset (Park et al.) to identify the clusters. mTEC (II) refers to AIRE-expressing mTECs in the reference paper, but AIRE expression is not observed here. Integrating the datasets from this study with the reference is necessary to determine whether the annotated clusters are closely related or overlapping.

2) The authors performed co-culture of induced TECs with thymocytes to validate the function of the induced TECs. After co-culture, do any of the TECs express AIRE? As the authors mentioned, and as is known, AIRE expression requires thymocyte-derived signals.

3) AIRE expression dynamics are considered important in mTEC subpopulations, including AIRE-negative mTECs, AIRE-expressing mTECs, and post-AIRE mTECs.

<6> While marker staining and flow cytometry data are provided, a more direct and comprehensive analysis using a golden standard flow cytometry analysis (for example, Ly51, UEA1, CD80, MHC-II (HLA-DR/DP/DQ)) to simultaneously assess both cTECs and mTECs would strengthen the analysis.

<7> Why were only DP thymocytes and mCherry+ iTECs used in the experiments shown in Figure 4? To fully assess iTEC functionality, all iTEC populations (mCherry-, mCherry low, and high) should be co-cultured with earlier-stage thymocytes, such as DN thymocytes. Or co-culture with hematopoietic stem and progenitor cells (HSPCs) is necessary.

1) Under similar conditions, co-culture of MS5-hDLL1 cells with CD34+CD3- HSPCs has been shown to generate mature CD4SP (~11%) and CD8SP (~25%) T cells (Seet et al., 2017, Nature Methods). The comparison presented in Figure 4c may not be entirely appropriate in light of these results.

2) During the development of DP thymocytes, CD69+ DP cells are known to migrate toward the medullary region, where interactions with mTECs are critical for further maturation into CD4 or CD8 SP T cells. This process involves negative selection, which plays a crucial role in establishing a self-tolerant T cell repertoire. However, the authors have primarily focused on positive selection. If only positive selection occurs, the TCR repertoire would likely become skewed, which does not appear to be the case in the results presented. Could the authors clarify whether negative selection is also achieved in this co-culture system? If not, what are the limitations, and if so, how is it facilitated?

3) The results of the co-culture were only analysed at day 14. However, including a day 0 control or additional time points would provide more comprehensive insights.

Reviewer #2

(Remarks to the Author)

In the present study, the authors established an in vitro system to develop human thymic epithelial cell populations from iPSCs in a 2D culture. The authors could show that the developed model recapitulates the thymic cell development and cell composition, despite some shortcoming with regards to the lack of AIRE expressing cells and the overlap in cell populations for TEC IV and V (Figure 6). Indicating that particularly late developmental programs might not be represented by the culture system as well as the earlier developmental stages. However, the detail to which the different cTEC and mTEC developmental populations did emerge in the iPSC-derived culture is promising and this assay will likely be highly beneficial to the research field, as current in vitro and ex vivo systems often lack a sufficient similarity to the in vivo context or do not allow culturing for a sufficient duration for functional studies during cell or tissue development.

Previous work done on iPSCs culturing to obtain TECs in culture should be referenced, and the results compared to

1) Immunol Cell Biol. 2011 Feb;89(2):314-21.

doi: 10.1038/icb.2010.96. Epub 2010 Aug 3.

Differentiation of induced pluripotent stem cells to thymic epithelial cells by phenotype

Yuta Inami 1, Tohru Yoshikai, Sachiko Ito, Naomi Nishio, Haruhiko Suzuki, Hidetoshi Sakurai, Ken-Ichi Isobe

2) Summary of studies performed in Front Immunol

. 2022 Jun 29;13:930963. doi: 10.3389/fimmu.2022.930963

Differentiation of Pluripotent Stem Cells Into Thymic Epithelial Cells and Generation of Thymic Organoids: Applications for Therapeutic Strategies Against APECED

Nathan Provin 1, Matthieu Giraud 1,*

Due to the emphasis on the method reported in this manuscript, the reported assay should be explained with regards to the used compounds and their function. What is the role of each of the compounds, CHIR99021, PI-103, LDN-193189, Activin A, A-83-01? Explanations should be added to the manuscript to facilitate a better understanding for the reader.

The authors further analyze the role of RA and Fgf8 during early development steps. The authors conclude in ms line 133 that from day 18 to day 28 no further increase in FOXN1 expression occurred, however the referenced Suppl Fig 1d showing

relative expression to day 18 seems to show 10 fold higher expression values? Showing D18 and D28 results next to each other would be helpful to read and interpret the data.

Introducing the gene regulatory network analysis and its purpose would be beneficial, similarly the term regulon (ms line 292-293).

The authors might explore a different display version for the results of the ligand-receptor interaction analysis (ms line 313-315, Suppl Fig. 6h). Which ligand-receptor pairs were identified and how many per population?

In the discussion the authors state in ms line 441-442 "our results indicate that early cTEC-like cells arise before mTEC lineage commitment and may still have multilineage potential until the downregulation of FOXP1 and cTEC markers", in addition to Ref 30 the authors should thoroughly cite and discuss previous work done on TEC progenitors and cTEC and mTEC origin. This has been a field of intensive study with opposing results and discussions in the field. Hence the manuscript would benefit from an embedding into those studies. Also, that mTECs derive from cTECs has been proposed and studied before and should be cited.

Version 1:

Reviewer comments:

Reviewer #1

(Remarks to the Author)

In this revised version, the authors have addressed most of the major concerns raised in the review, primarily through new experimental data. These additions have helped clarify aspects of the manuscript that were previously ambiguous. Furthermore, the authors provided generally well-reasoned and convincing responses to the comments. As a result, the manuscript has been meaningfully improved and now presents a more solid and coherent body of work. The revised study is expected to contribute to the field of iPSC-based in vitro thymic epithelial cell modeling by offering a useful new model that others can build upon.

Reviewer #3

(Remarks to the Author)

I wish to clarify that I'm not an original reviewer but that I rather mediate for the missing Reviewer 2 at this stage of the manuscript evaluation process (revision stage).

I have looked at the comments and remarks that were made by both reviewers and it appeared that both reviewers showed enthusiasm for the work already upon initial submission. One particular issue that was raised by both reviewers was the lack of AIRE expressing TECs in this newly designed methodology that aims to generate various human TEC populations from human iPSCs. I agree with both reviewers that this is indeed an important point (but also that this work even without this would provide a substantial advancement for the field as also mentioned). The authors have now addressed this by performing new coculture experiments with thymocytes to show that AIRE expressing cells can be detected in their iPSC-derived TECs, in line with the knowledge that differentiating thymocytes indeed are needed to induce TEC maturation into AIRE expressing subsets through RANK-RANKL interactions. These new results, which are now described in a whole new section in the revised manuscript, not only address these concerns but also significantly strengthen the paper and further solidify the proposed methodology to generate human TECs from iPSCs.

In addition to this, I believe that the authors have adequately addressed the other points that were raised by Reviewer 2, including incorporation of relevant literature, explanation of the role of various compounds that are used in the protocol and some other requested clarifications (terminology and figure improvement).

As such, I believe that the authors have adequately revised their manuscript.

Point-by-point response to reviewer comments

We thank the reviewers for their constructive and insightful feedback on our manuscript. Please see our responses below, shown in blue, and the changes in the manuscript, shown in yellow highlights.

Reviewer #1 (Remarks to the Author):

The authors established a human iPSC-based model to study thymus organogenesis using a simple 2D culture system. The model demonstrates that controlled RA treatment can guide PE to adopt a thymic fate, generating FOXP3⁺ cells. Additionally, the system produces diverse TEC-like populations through self-directed differentiation over a period of 130 days. The induced TEC-like cells also support in vitro generation of naïve T cells. Single-cell gene expression profiles further illustrate that this model generates diverse TEC-like populations, including cortical and medullary subtypes comparable to primary human TECs.

The experimental design is well-constructed, and the conclusions drawn from the data appear appropriate. However, several major and minor issues remain that should be addressed to strengthen the manuscript's claims.

<1> In figure 1, a key finding may be the induction of PE through the addition of FGF8 and retinoid acid, allowing cells progress from the PE stage through self-directed differentiation toward TEP and subsequently into TECs by culturing over 130 days.

While this strategy is effective, it is worth noting that similar concentrations of RA and FGF8 have been previously employed in studies using ES cells to drive differentiation to the PE stage, which may slightly reduce the novelty of this approach. However, a notable strength of this study is the extended period of over 130 days for self-directed differentiation beyond the PE stage, allowing the maturation of diverse TEC-like populations in vitro.

1) Based on this, an explanation (using data) of how self-directed differentiation can be achieved in vitro might be more important than emphasizing controlled retinoic acid treatment.

We agree that self-directed differentiation is indeed an important aspect of our induction system. We have performed a range of experiments to further emphasize this point and present this data in Fig. 2g and Supplementary Fig. 1e, 3c, 4d, 9b. Details of these new results are in our response to each of the points below.

2) At later time point, diverse TEC populations were observed, including cTECs and mTECs. The differentiation of these populations may require various signaling pathways, such as

NOTCH signaling, LTBR signaling, and NF-kB signaling. During the differentiation process (PE - TEP - TEC), were these signalling-related molecules observed in a time dependent manner?

To examine these signaling pathways during the induction, we looked at the mRNA expression of several of their crucial components and downstream genes. We find that both NOTCH and NF-kB signaling molecules are indeed increased in a time-dependent manner from day 0 to 28 (Supplementary Fig. 1e). After day 28, NOTCH remains stable, while NF-kB is further increased especially in the mCherry^{high} population (Supplementary Fig. 3c). Since LTBR is known to be upstream of NF-kB and is consistently expressed in all populations in our system (Supplementary Fig. 4d), it may mediate the increase of NF-kB signaling that we see over time. The high expression of both NOTCH and NF-kB signaling molecules without the addition of any signaling modulators after day 18 underscores that the differentiation proceeds in a self-directed manner. These results are consistent with previous murine studies (ref. #73-76) showing the involvement of these pathways in multiple stages of TEC development, as well as with the previously reported human primary TEC dataset by Park et al. (ref. #30) showing broad expression of genes of these signaling pathways in both cTECs and mTECs (see our reanalysis of their dataset below).

We have added the following explanations to the Results section describing our findings:

- Under subheading “**Induction of TEP-like cells through RA-based endodermal patterning**”: “Indeed, signaling molecules of pathways involved in both early and later stages of TEC development, such as NOTCH^{73,74} and NF-kB^{75,76}, automatically increased over time without further signaling manipulation, suggesting possible self-directed differentiation after 3rd PP specification (Supplementary Fig. 1e).”
- Under subheading “**Self-directed differentiation of cTEC- and mTEC-like populations**”: “Both NOTCH and NF-kB signaling genes remained highly expressed after day 28 in all populations, with NF-kB being especially prominent in the

mCherry^{high} population, [...] (Supplementary Fig. 3c).”

3) Figure 1i is not clear to understand. Proper legend in the figure should be added.

We apologize for the lack of clarity in Figure 1i. To facilitate understanding, we have added the specific signaling modulators and their concentrations to the figure legend rather than a separate supplementary table. The legend now includes the following information:

- “The signaling modulators and concentrations are as follows: WNT inhibition, 0.2 μ M, 1 μ M, 5 μ M XAV939; WNT activation, 0.4 μ M, 2 μ M, 10 μ M CHIR99021; HH inhibition, 0.2 μ M, 1 μ M, 5 μ M Cyclopamine; HH activation, 20 nM, 100 nM, 500 nM SAG; BMP inhibition, 40 nM, 200 nM, 1 μ M LDN-193189; BMP activation, 4 ng/ml, 20 ng/ml, 100 ng/ml BMP4; NOTCH inhibition, 0.1 μ M, 0.5 μ M, 2.5 μ M RO4929097; NOTCH activation, 0.2 mM, 1 mM, 5 mM Valproic acid; FGF inhibition, 40 nM, 200 nM, 1 μ M BGJ398; FGF activation, 2 ng/ml, 10 ng/ml, 50 ng/ml FGF8.”

Furthermore, we have incorporated the underlying data originally shown in separate supplementary tables into a single Source Data file in Microsoft Excel format, arranged into sheets ordered by figure. All values of Figure 1i, as well as other figures, are now shown in this file. The untreated control (-) row is now also shown within the figure itself, similarly to Fig. 1g.

<2> In the thymus, the cortical and medullary regions are distinct, with specific TEC subpopulations observed in each region. In this model, are both cortical and medullary regions observed? In figure 2 and supplementary figure 3/4, each marker is visualised together with FoxN1, only. Co-staining with general cTEC and mTEC markers simultaneously is required to determine whether this model also recapitulate the spatial distribution of cTECs and mTECs.

It is mentioned that “By day 133, 3D-like nodules up to 100 μ m in height were observed within mCherry+ colonies.”. What do these high 3D-like nodules represent? How are the expression of cTEC or mTEC markers associated with the formation of these 3D-like nodules?

To examine whether the spatial separation of cortical and medullary regions in the thymus is recapitulated in our model, we performed simultaneous immunostaining of PSMB11, specific for cTECs, and KRT5, specific for mTECs (Fig. 2g). We find that cTEC- and mTEC-like areas are indeed mostly separate, although some overlap is also seen, possibly reflecting the cTEC/mTEC junctional areas in the thymus. It is largely within these overlapping areas that we observed the 3D-like nodules (Supplementary Video 2), where both co-staining cells and mosaic-like patterns of PSMB11⁺ and KRT5⁺ cell clusters are found. Interestingly, many of these nodules have inner areas consisting of vimentin-expressing mesenchyme-like cells, with only the outer layer being FOXN1⁺, likely leading

to close epithelial-mesenchymal interactions in these structures (Supplementary Fig. 9b, c and Supplementary Video 3). *In vivo*, mesenchyme contributes to the establishment of thymic niches, with fibroblastic cells being abundant at the corticomedullary junction – one of the niches of TEC progenitor populations (ref. #75, #86, #87, #112, #140). In light of these studies, together with our scRNA-seq results in Fig. 5-7 indicating that *PSMB11*⁺*KRT5*⁺ cells may include precursors to both the cTEC and mTEC I lineages, we speculate that the *PSMB11*/*KRT5* overlapping areas and mesenchyme-rich nodules may represent such a distinct niche akin to the corticomedullary junction. We describe our findings in the Results section as follows:

- Under subheading “**Self-directed differentiation of cTEC- and mTEC-like populations**”: “By day 133, *PSMB11*⁺ cTEC-like areas and *KRT5*⁺ mTEC-like areas were largely spatially separated with some overlapping junctional areas where 3D-like nodules formed, implying the presence of distinct niches similar to the thymus^{75,86,87} (Fig. 2g and Supplementary Video 2).”
- Under subheading “**iTECs consist of heterogeneous populations resembling primary TECs**”: “[...] epithelial-mesenchymal interactions that may contribute to the self-directed differentiation in our system. These mesenchyme-like cells, marked by *VIM*, are present by day 28 and closely associate with *mCherry*⁺ colonies by day 133, with localized accumulations of these cells below the *mCherry*⁺ layer being observed in 3D-like nodules (Supplementary Fig. 9b, c and Supplementary Video 3).”

<3> It could be more impactful if gene editing were used to generate a lineage tracing system for *FOXN1* (for example, using *FOXN1* lineage with GFP and *FOXN1* expression with *mCherry*). This would further support the author’s claim regarding the mimetic mTECs (from lines 383 to 387), suggesting that some mTEC differentiation occurs independently of *FOXN1* expression.

We agree that a lineage tracing system would be an excellent method to confirm or deny the dependence of different TEC populations on *FOXN1*. Indeed, we expect that the creation of such a system and its use to discover the origins of each mimetic mTEC subpopulation *in vitro* will be one of the major applications of this new induction system. Therefore, we believe that this would merit its own future publication, especially in light of the results we show in response to comment <5>. Since it is difficult to establish such a lineage tracing system in a short timeframe, we instead performed a timecourse of mRNA expression analysis by qPCR in each sorted population to provide further evidence of potential *FOXN1*-independent development (Supplementary Fig. 10f). We looked at genes we identified using scRNA-seq analysis to be highly specific to the path leading to the

mTEC III cluster – *FOXA1* and *FOXE1* – and found that their transient expression was indeed seen almost exclusively in the mCherry⁺ population. Secretory markers *PSCA* and *CXCL17* were also seen mostly in the mCherry⁺ population, with some expression also in mCherry^{low} by day 133, possibly reflecting the differentiation of such mimetic mTEC-like cells from both the mTEC II and III clusters. However, since *FOXN1* could still be transiently expressed in these mTEC III-fated cells at timepoints we did not examine, this evidence is not fully conclusive and will need to be supported by lineage tracing in the future. We describe these findings and their limitations in the Results section under the subheading “**mTEC-like lineages diverge at varying stages of cTEC development**” as follows:

- “The transient expression of *FOXA1* and *FOXE1*, as well as the subsequent upregulation of secretory markers *PSCA* and *CXCL17*, was found almost exclusively in the mCherry⁺ population, suggesting that this lineage may have the potential to develop independently of *FOXN1*, although this will require further confirmation by lineage tracing (Supplementary Fig. 10f).”

<4> After iPSCs are induced into TEPs, are there any proliferating subpopulations during further differentiation process? Can TEPs regenerate or self-renew, as previously described?

To investigate whether any proliferating subpopulations remain after the induction into TEP-like cells, we stained for Ki-67 on both day 28 and 133 (Supplementary Fig. 3d). Although fewer in number compared to day 28, some Ki-67⁺ cells were detected even on day 133, indicating that there is indeed a self-renewing progenitor-like population even after long-term culture. Further evidence for this is seen in the expression of *DLK2*, a recently identified marker of postnatal progenitors, from day 57 onward in the mCherry^{high} and mCherry^{low} populations (Supplementary Fig. 3c). From these results, and considering that TECs have an estimated lifespan of only about 2 weeks (Gray DH, Seach N, Ueno T, Milton MK, Liston A, Lew AM, Goodnow CC, Boyd RL. Developmental kinetics, turnover, and stimulatory capacity of thymic epithelial cells. *Blood*. 2006 Dec 1;108(12):3777-85. doi: 10.1182/blood-2006-02-004531.), we assume that there is likely a slow but continuous turnover of iTECs in our culture. We describe these findings in the Results section under the subheading “**Self-directed differentiation of cTEC- and mTEC-like populations**” as follows:

- “[...] and postnatal TEC progenitor marker^{29,30} *DLK2* being expressed by mCherry^{high} and mCherry^{low} cells in the later stages (Supplementary Fig. 3c). Ki-67⁺ cells were detected even on day 133, suggesting the development and long-term maintenance of a progenitor-like population in our system (Supplementary Fig. 3d).”

<5> The absence of AIRE expression at all time points raises a significant concern and weakness in the study, as functional TEC differentiation should inherently include AIRE-expressing mTECs.

We agree that *AIRE*-expressing mTECs are an essential subset of TECs and their absence in our system is a notable shortcoming, as also pointed out by Reviewer #2. Therefore, as suggested below, we decided to examine the potential for *AIRE* expression by thymocyte co-culture. As a result, we find that iTECs are indeed able to upregulate *AIRE* expression after 2 weeks of co-culture, indicating successful differentiation of *AIRE*⁺ mTEC-like cells (Fig. 8a, b). To further investigate this important population, we then performed scRNA-seq of the co-cultured cells. We detected the *AIRE*⁺ iTECs as a subpopulation within the mTEC II cluster, directly connected to clusters of likely post-*AIRE* neuronal mimetic mTEC-like cells, so we further assessed their dynamics using RNA velocity and pseudotime analysis (Fig. 8c-i and Supplementary Fig. 11a-g). Further details are provided in the responses to the comments below. Since the capability to model the development of *AIRE*⁺ mTECs using iPSCs *in vitro* has now become a significant strength of our system, we have created a new subheading named “iTECs model *AIRE*⁺ mTEC differentiation in thymocyte co-culture” in the Results section, where we describe these analyses.

1) In the scRNA-sequence dataset, the authors used the reference dataset (Park et al.) to identify the clusters. mTEC (II) refers to AIRE-expressing mTECs in the reference paper, but AIRE expression is not observed here. Integrating the datasets from this study with the reference is necessary to determine whether the annotated clusters are closely related or overlapping.

By integrating our scRNA-seq dataset with that from Park et al., we find that our cTEC, mTEC I, mTEC II, mTEC III, and neuronal mimetic mTEC-like clusters indeed mostly overlap with their counterparts in the reference dataset, similarly to the co-clustering we see with our primary thymic stromal dataset (Fig. 8d and Supplementary Fig. 11b). The mTEC II cluster is, however, also partially adjacent or co-clustering with other clusters, likely due to its relative immaturity and heterogeneity (Supplementary Fig. 11d-f), as the mTEC(II) cluster from Park et al. is represented mostly by late fetal and postnatal samples (Park et al. Fig. S4B). Therefore, our mTEC II cells are partially similar, but not completely identical, to their primary counterparts seen in the later stages.

2) The authors performed co-culture of induced TECs with thymocytes to validate the function of the induced TECs. After co-culture, do any of the TECs express AIRE? As the authors mentioned, and as is known, AIRE expression requires thymocyte-derived signals.

We are thankful for this important suggestion. *AIRE* is indeed detected after thymocyte co-culture both using qPCR and scRNA-seq, demonstrating the potential for *AIRE*⁺ mTEC

differentiation in our system (Fig. 8a-f).

3) *AIRE* expression dynamics are considered important in mTEC subpopulations, including *AIRE*-negative mTECs, *AIRE*-expressing mTECs, and post-*AIRE* mTECs.

To assess *AIRE* expression dynamics, we performed RNA velocity and pseudotime analysis of our co-cultured iTECs and find its transient expression within the mTEC II cluster, followed by upregulation of neuronal markers (Fig. 8g-i). The expression of *AIRE* is concurrent with several crucial markers of *AIRE*⁺ mTECs, including *INSM1*, a possible regulator of *AIRE* and neuronal mimetic mTEC differentiation (ref. #129). During this process, progenitor markers are downregulated, and, by the end point, markers of differentiated neurons and neuroendocrine cells are detected (Fig. 8i). This closely reflects the differentiation of neuronal mimetic mTECs from progenitors through the *AIRE*⁺ stage, which, crucially, has hitherto not yet been achieved with pluripotent stem cells. Although these neuronal mimetic mTEC-like cells were the only post-*AIRE* population we detected at this stage, these results demonstrate the potential for *AIRE*-dependent mimetic mTEC differentiation *in vitro*, and we aim to further improve and apply our system to the study of mimetic mTEC development in future publications.

We describe the results above in the “**iTECs model *AIRE*⁺ mTEC differentiation in thymocyte co-culture**” Results section as follows:

- “Since *AIRE* was not expressed at any time point during the induction, we next examined whether iTECs have the potential for *AIRE*⁺ mTEC differentiation by co-culturing them with thymocytes, known to promote mTEC development^{81,82}. After two weeks of organoid co-culture of day 37 mCherry⁺ iTECs with DP thymocytes, *AIRE* was significantly upregulated in the mCherry⁺ fraction, reaching 6% of primary TECs (Fig. 8a, b). scRNA-seq of the organoids revealed diverse TEC-like populations similar to the non-co-cultured iTECs, with clusters of cTEC, mTEC I, mTEC II, and mTEC III that co-cluster closely with their primary counterparts (Fig. 8c, d, and Supplementary Fig. 11a, b). Mesenchyme-like clusters were also detected with partial co-clustering of primary mTECs and induced mTEC I from D133, as well as their respective Mes clusters, implying that mCherry⁺ iTECs may partially undergo EMT in organoid co-culture, similarly to TECs acquiring EMT signatures during aging *in vivo*^{125,126} (Fig. 8c, d, and Supplementary Fig. 11a, c). In addition, small lymphocyte clusters including one containing *FOXP3*⁺*IL2RA*⁺ cells were present, indicating possible iTEC functionality of inducing regulatory T cells (Tregs) from the DP stage (Fig. 8c, e and Supplementary Fig. 11c). *AIRE* expression was observed within a section of the mTEC II cluster directly connected to clusters co-expressing neuronal markers and *EPCAM*, reminiscent of neuronal mimetic mTECs (Fig. 8e, f). These

clusters, assigned as mTEC-mim I-III, co-clustered closely with the mTEC-mim cluster in our dataset of human primary thymic stroma, as well as the TEC(neuro) cluster in the dataset of human thymic epithelium by Park et al.³⁰ (Fig. 8d and Supplementary Fig. 11b).

RNA velocity analysis, with the inferred latent time representing the cell's internal clock¹²⁷, indicates the likely origin of these neuronal mimetic mTEC-like cells to be a subset within the mTEC II cluster expressing various keratins (Fig. 8g, h, and Supplementary Fig. 11d-f). The expression of keratins and *BCAM*, reflective of progenitors³¹, decreased across pseudotime, while recently identified^{30,128} markers of *AIRE*⁺ mTECs, such as *NTHL1*, *CALML3*, and *INSM1*, increased either transiently or permanently (Fig. 8i). *ELF3*, *EHF*, and *GRHL1* were expressed broadly within the mTEC II and the secretory mimetic mTEC III clusters independently of *AIRE* expression (Supplementary Fig. 11g). *AIRE*⁺ mTEC markers peaked nearly concurrently with those of early neuronal differentiation, including *ASCL1* and *NEUROG1*, possibly driven by *INSM1*¹²⁹ (Fig. 8i). Thereafter, genes abundantly expressed in neurons, such as *STMN2* and *RTN1*, increased especially in mTEC-mim II, and markers more specific to differentiated neuronal subtypes, such as cerebellar Purkinje cells (*PCP4*), as well as glia (*METRN*) and cochlear hair cells (*ATOH1*), appeared in mTEC-mim III (Fig. 8i). Together, these results show that, upon co-culture with thymocytes, our iTECs are capable of recapitulating the dynamics of *AIRE*⁺ mTEC and post-*AIRE* neuronal mimetic mTEC development.”

Due to the importance of these results, we have also amended the Abstract, Introduction, and Discussion sections to include the following:

- Abstract: “Upon thymocyte co-culture, induced TECs [...] further develop into *AIRE*⁺ and mimetic TEC subpopulations.”
- Introduction: “[...] through which we further obtain *AIRE*⁺ and post-*AIRE* populations.”
- Discussion: “Moreover, in an important advance for PSC-based modeling of the thymus, our iTECs are able to differentiate into *AIRE*⁺ mTEC-like cells and recapitulate *AIRE*-dependent neuronal mimetic mTEC development in thymocyte co-culture, indicating that they are able to respond to signals from thymocytes for further differentiation up to the terminal stages similarly to primary TECs. [...] After co-culture with thymocytes, we were able to detect *AIRE*⁺ mTEC-like cells, possibly corresponding to the earliest *AIRE*-expressing TECs during week 13 of fetal development⁴. [...] While the heterogeneity of the terminally differentiated mimetic population is not yet fully represented in our system, with most mimetic mTEC-like

cells by day 133 having a secretory or keratinized phenotype, and all likely post-*AIRE* cells in thymocyte co-culture assuming a neuronal or neuroendocrine phenotype, further improvements in modeling the lymphostromal crosstalk of the thymus *in vitro* may yet yield other subtypes of mimetic mTECs and allow further exploration of their origins in humans.”

<6> While marker staining and flow cytometry data are provided, a more direct and comprehensive analysis using a golden standard flow cytometry analysis (for example, Ly51, UEA1, CD80, MHC-II (HLA-DR/DP/DQ)) to simultaneously assess both cTECs and mTECs would strengthen the analysis.

We agree that the suggested flow cytometry panel is indeed an excellent way to assess cTECs and mTECs, as seen in numerous murine studies. Unfortunately, however, markers of cTECs and mTECs are much less well defined in humans, where many of these murine markers either do not stain at all, stain only in a small subset of cells, or show drastically different staining patterns altogether. As an example, UEA1, used to reliably identify mTECs in mice, stains in only very few mTECs in the postnatal human thymus, nearly all TECs in the fetal human thymus at 7 weeks, and also some non-epithelial cells (see the human studies in ref. #4 and Haunerding et al. and Villegas et al. below).

- Haunerding V, Moccia MD, Opitz L, Vavassori S, Dave H, Hauri-Hohl MM. Novel Combination of Surface Markers for the Reliable and Comprehensive Identification of Human Thymic Epithelial Cells by Flow Cytometry: Quantitation and Transcriptional Characterization of Thymic Stroma in a Pediatric Cohort. *Front Immunol.* 2021 Sep 30;12:740047. doi: 10.3389/fimmu.2021.740047.
- Villegas JA, Gradolatto A, Truffault F, Roussin R, Berrih-Aknin S, Le Panse R, Dragin N. Cultured Human Thymic-Derived Cells Display Medullary Thymic Epithelial Cell Phenotype and Functionality. *Front Immunol.* 2018 Jul 23;9:1663. doi: 10.3389/fimmu.2018.01663.

Haunerding et al. also tested CD80, used to identify mature mTECs in mice, but could not detect its expression at the protein level in human TECs, and the mRNA expression was very low. In the scRNA-seq dataset by Park et al., *CD80* is only seen in a small subset of the *AIRE*⁺ population (see our reanalysis of this dataset below).

For cTECs, CD205 (LY75) has emerged as a reliable marker (see the recent human studies by Ragazzini et al. and Campinoti et al. – ref. #31 and #34). Its expression patterns are similar to LY51 (ENPEP) (Park et al. dataset reanalysis above). Hence, we used CD205 to identify cTECs in our system.

In the absence of a clear mTEC marker in humans, Ragazzini et al. and Campinoti et al. used CD205 together with EPCAM, which marks all TECs but is generally higher in mTECs, to define cTECs as CD205⁺EPCAM^{low} and mTECs as CD205⁻EPCAM^{high}. Campinoti et al. also show that the expression of *FOXP1* is an order of magnitude higher in cTECs compared to mTECs. In light of these results, we chose CD205 in conjunction with *FOXP1*, easily detectable using mCherry in our reporter iPSC lines, to identify cTECs as CD205^{high}*FOXP1*^{high} and mTECs as CD205⁻*FOXP1*^{low}, with HLA-DR/DQ further defining the maturity of these populations. We believe that, given the current lack of a reliable human mTEC surface marker, as well as the strategies used in previous human TEC studies, our flow cytometry closely reflects the current golden standard for human cTEC/mTEC identification.

<7> Why were only DP thymocytes and mCherry⁺ iTECs used in the experiments shown in Figure 4? To fully assess iTEC functionality, all iTEC populations (mCherry⁻, mCherry low, and high) should be co-cultured with earlier-stage thymocytes, such as DN thymocytes. Or co-culture with hematopoietic stem and progenitor cells (HSPCs) is necessary.

The reason we used pre-selection DP thymocytes and mCherry⁺ iTECs in the experiments of Figure 4 was to directly demonstrate the capability of our iTECs to support positive selection, a crucial function of cTECs that has been difficult to recapitulate using other *in vitro* systems such as NOTCH ligand-expressing cell lines. We have now clarified this in

our manuscript (see below). However, we agree that assessing iTEC functionality more broadly is important to provide more comprehensive insights. Therefore, as suggested, we have performed additional co-culture experiments using mCherry⁻ iTECs, as well as DN thymocytes, and we show these results in Supplementary Fig. 6a, 6e, and 7a-e. Although there is a higher propensity toward TCR $\gamma\delta$ and CD8 $\alpha\alpha$ phenotypes when starting from the DN stage, we find that mCherry⁺ iTECs are indeed also capable of supporting the development of conventional TCR $\alpha\beta$ ⁺ thymocytes with CD4/8 DP and SP phenotypes from such earlier-stage thymocytes (Supplementary Fig. 7a-d). In contrast, mCherry⁻ iTECs could neither maintain thymocytes, nor support their development, nor engage in positive selection, demonstrating that these are functions exclusive to the mCherry⁺ population (Supplementary Fig. 6e, 7e). Please note that we do not further divide the mCherry⁺ population into mCherry^{high} and mCherry^{low} for these co-culture experiments because a clear split between these fractions is only seen on day 57 and later, and the iTECs used for co-culture are day 37, where the distinction between “high” and “low” would be arbitrary. We describe our additional co-culture experiments and added clarification in the Results section under the subheading “**Induced TECs are mature and capable of positive selection *in vitro***” as follows:

- “We then proceeded to assess the functionality of our iTECs. Up to now, various systems for *in vitro* T cell production have been developed, including DLL1-expressing stromal cell lines and feeder-free DLL4 microbead systems, but while they are highly adept at generating CD4⁺CD8⁺ double-positive (DP) T lineage cells, efficient induction of mature single-positive (SP) cells, especially CD4⁺ SP cells, remains challenging^{40,92–94}. Therefore, we tested the capability of our iTECs to engage in positive selection, a crucial event in the development of SP cells orchestrated by cTECs in the thymus^{1,2,7}. We sorted day 37 mCherry⁺ iTECs and co-cultured them with pre-selection (CD69⁻) DP thymocytes from pediatric human thymi in 3D organoids at the air-liquid interface, similarly to a previous report using the murine stromal MS5-hDLL1 cell line⁹², with the same cell line used as control (Fig. 4a, b and Supplementary Fig. 6a, b). [...]

When co-culturing mCherry⁺ iTECs with earlier-stage CD4⁺CD8⁻ double-negative (DN) thymocytes, we found a greater propensity toward innate-like TCR $\gamma\delta$ and CD8 $\alpha\alpha$ phenotypes, likely in part due to the inherent heterogeneity of DN cells⁹⁸, but conventional CD3⁺TCR $\alpha\beta$ ⁺ thymocytes including 45% DP, 8% CD4⁺ SP, and 42% CD8⁺ SP cells were also generated (Supplementary Fig. 7a-d). In contrast, mCherry⁻ iTECs could neither induce SP thymocytes nor maintain DP or DN thymocytes in co-culture (Supplementary Fig. 6e, 7e). Together, these results demonstrate the

capability of our iTECs, specifically the mCherry⁺ population, to support both early T cell development and later positive selection of SP cells with diverse TCR repertoires.”

1) Under similar conditions, co-culture of MS5-hDLL1 cells with CD34⁺CD3⁻ HSPCs has been shown to generate mature CD4SP (~11%) and CD8SP (~25%) T cells (Seet et al., 2017, Nature Methods). The comparison presented in Figure 4c may not be entirely appropriate in light of these results.

Seet et al. indeed show that co-culture of MS5-hDLL1 with CD34⁺CD3⁻ HSPCs generates mature CD4 and CD8 SP T cells. However, this process takes a very long time. Mature, naïve SP cells arise only after 10-12 weeks of co-culture with MS5-hDLL1 (Fig. 3a-c and Supplementary Fig. 6a in Seet et al.). Considering that CD4⁺CD8⁺ DP cells already arise by week 2-3 under these conditions (Fig. 1b in Seet et al.), this means that this process takes up to 2 months from the DP stage. Indeed, Supplementary Fig. 6a in Seet et al. shows that by week 6, 83% of CD3⁺TCRαβ⁺ cells are still CD4⁺CD8⁺ DP. This is very similar to our results in Figure 4c, where MS5-hDLL1 was co-cultured with DP thymocytes for 2 weeks, after which 91% of CD3⁺TCRαβ⁺ cells are still CD4⁺CD8⁺ DP. The results of both our study and Seet et al. show that MS5-hDLL1 struggles with positive selection in the short term. Hence, we believe that our comparison in Figure 4c is appropriate. To clarify this point, we have added the following to the Results section under the subheading “**Induced TECs are mature and capable of positive selection *in vitro***”:

- “After two weeks of organoid co-culture, MS5-hDLL1 still had 91% of the CD3⁺TCRαβ⁺ population remaining as CD4⁺CD8⁺ DP, consistent with the previously reported long co-culture periods required to produce mature SP cells⁹², while iTECs led to CD4⁺ and CD8⁺ SP phenotypes in respectively 47% and 21% of CD3⁺TCRαβ⁺ (Fig. 4c, d).”

2) During the development of DP thymocytes, CD69⁺ DP cells are known to migrate toward the medullary region, where interactions with mTECs are critical for further maturation into CD4 or CD8 SP T cells. This process involves negative selection, which plays a crucial role in establishing a self-tolerant T cell repertoire. However, the authors have primarily focused on positive selection. If only positive selection occurs, the TCR repertoire would likely become skewed, which does not appear to be the case in the results presented. Could the authors clarify whether negative selection is also achieved in this co-culture system? If not, what are the limitations, and if so, how is it facilitated?

The *in vitro* recapitulation of not only positive but also negative selection in an iPSC-based model is indeed an issue of high interest for both basic immunological research and clinical applications. Although we do not observe noticeable skewing of the TCR repertoire, indicating the selection of diverse clones, it is difficult to estimate the degree of negative selection on the basis of the TCR repertoire alone, as it is known that the biases between

pre- and post-selection cells are small, no specific TCR sequences are suppressed by negative selection, not all self-reactive clones are deleted, and self-reactivity can only be somewhat inferred from TCR sequences with computational modeling:

- Camaglia F, Ryvkin A, Greenstein E, Reich-Zeliger S, Chain B, Mora T, Walczak AM, Friedman N. Quantifying changes in the T cell receptor repertoire during thymic development. *Elife*. 2023 Jan 20;12:e81622. doi: 10.7554/eLife.81622.
- Textor J, Buytenhuijs F, Rogers D, Gauthier ÈM, Sultan S, Wortel IMN, Kalies K, Fähnrich A, Pagel R, Melichar HJ, Westermann J, Mandl JN. Machine learning analysis of the T cell receptor repertoire identifies sequence features of self-reactivity. *Cell Syst*. 2023 Dec 20;14(12):1059-1073.e5. doi: 10.1016/j.cels.2023.11.004.

Given that we detected both *AIRE*⁺ cells – essential for negative selection – and Treg-like cells in our scRNA-seq dataset after the co-culture of iTECs with DP thymocytes, it is indeed possible that iTECs have the potential to engage in negative selection and promote central tolerance. We have added a mention of this possibility to the Discussion:

- “The emergence of *AIRE*⁺ and *FOXP3*⁺ cells after thymocyte co-culture also raises the possibility that iTECs could support negative selection and Treg generation, which will be an important avenue to explore for applications to autoimmune diseases.”

3) The results of the co-culture were only analysed at day 14. However, including a day 0 control or additional time points would provide more comprehensive insights.

To address this point, we have added a day 0 control for both the DP and DN thymocyte co-culture experiments, showing that the sorted DP and DN populations on day 0 do indeed have the described phenotype and are highly pure, substantiating our results of iTEC functionality (Supplementary Fig. 6a and 7b).

Reviewer #2 (Remarks to the Author):

In the present study, the authors established an in vitro system to develop human thymic epithelial cell populations from iPSCs in a 2D culture. The authors could show that the developed model recapitulates the thymic cell development and cell composition, despite some shortcoming with regards to the lack of *AIRE* expressing cells and the overlap in cell populations for TEC IV and V (Figure 6). Indicating that particularly late developmental programs might not be represented by the culture system as well as the earlier developmental stages. However, the detail to which the different cTEC and mTEC developmental populations did emerge in the iPSC-derived culture is promising and this assay will likely be highly beneficial to the research field, as current in vitro and ex vivo

systems often lack a sufficient similarity to the *in vivo* context or do not allow culturing for a sufficient duration for functional studies during cell or tissue development.

Previous work done on iPSCs culturing to obtain TECs in culture should be referenced, and the results compared to

1) *Immunol Cell Biol.* 2011 Feb;89(2):314-21.

doi: 10.1038/icb.2010.96. Epub 2010 Aug 3.

Differentiation of induced pluripotent stem cells to thymic epithelial cells by phenotype

Yuta Inami 1 , Tohru Yoshikai, Sachiko Ito, Naomi Nishio, Haruhiko Suzuki, Hidetoshi Sakurai, Ken-Ichi Isobe

2) Summary of studies performed in *Front Immunol*

. 2022 Jun 29;13:930963. doi: 10.3389/fimmu.2022.930963

Differentiation of Pluripotent Stem Cells Into Thymic Epithelial Cells and Generation of Thymic Organoids: Applications for Therapeutic Strategies Against APECED

Nathan Provin 1, Matthieu Giraud 1,*

For better comparison with previous works, we have added citations of all studies given above, including those summarized in 2), both in the Introduction and Discussion.

- Introduction: “In the past decade, several reports have shown successful induction of *FOXP1*⁺ TEP-like cells able to contribute to the reconstitution of thymic function in immunodeficient mice upon *in vivo* transplantation^{36–48}.”
- Discussion: “While previous PSC-based protocols have required transplantation, complex 3D co-cultures, or decellularized scaffolds for TEC induction^{36–48}, our system can produce mature iTECs using simple, chemically defined 2D culture, allowing us to observe their development in a stable, easily reproducible environment.”

Furthermore, we have clarified the advances in this manuscript to not only mention the induction of heterogeneous cTEC and mTEC populations, but also the stability of *FOXP1* and *MHCII* expression in long-term 2D culture, as well as the differentiation of *AIRE*⁺ and post-*AIRE* mTEC populations in thymocyte co-culture, in Discussion paragraph 1:

- “[...], our iTECs stably express *FOXP1* and MHC class II molecules even after 133 days in 2D conditions, indicating that these crucial signals were successfully recapitulated and maintained in our culture. This long-term stability overcomes the limitation of previous induction protocols maintaining TEC-like cells for only up to 30 days with low expression of these markers in 2D when compared to primary TECs^{36–}

⁴⁸. Moreover, in an important advance for PSC-based modeling of the thymus, our iTECs are able to differentiate into *AIRE*⁺ mTEC-like cells and recapitulate *AIRE*-dependent neuronal mimetic mTEC development in thymocyte co-culture, indicating that they are able to respond to signals from thymocytes for further differentiation up to the terminal stages similarly to primary TECs. Together, these capabilities demonstrate the fidelity of our system and provide a unique opportunity to shed light on developmental processes that have otherwise been exceedingly difficult to observe.”

Due to the emphasis on the method reported in this manuscript, the reported assay should be explained with regards to the used compounds and their function. What is the role of each of the compounds, CHIR99021, PI-103, LDN-193189, Activin A, A-83-01? Explanations should be added to the manuscript to facilitate a better understanding for the reader.

For greater ease of understanding, we have added explanations of the roles and functions of each of the compounds, including several references, to the Methods section under the subheading “**TEC induction from iPSCs**” as follows:

- “Activin A is known to drive pluripotent stem cells toward the PS and its derivatives when PI3K signaling is suppressed¹⁵¹ and WNT signaling is active⁴⁹. PI-103, a potent PI3K inhibitor, and CHIR99021, a GSK3 inhibitor promoting WNT activation, are capable of effectively specifying the PS⁴⁹. On day 1, the medium was changed to CDM2 with 100 ng/ml Activin A and 250 nM LDN-193189 (Reprocell) to induce definitive endoderm (DE). Since BMP signaling at this stage leads to the formation of mesoderm¹⁵², BMP inhibitor LDN-193189 is used to specifically induce DE⁴⁹. On day 3, the medium was changed to CDM2 with 250 nM LDN-193189 and 1 μ M A-83-01 (Tocris) for AFE induction. The AFE fate is promoted by the combined inhibition of BMP and TGF β signaling¹⁵³, with A-83-01 being a potent inhibitor of the TGF β pathway effective at specifying AFE⁴⁹.

Next, to induce pharyngeal endoderm (PE) from AFE, the medium was changed to CDM2 supplemented with 150 nM all-trans retinoic acid (RA, Sigma-Aldrich) and 50 ng/ml FGF8 (R&D Systems) from day 7 to 17. RA is responsible for the patterning of the posterior pharyngeal pouches⁵⁷⁻⁵⁹ and FGF8 promotes the normal development and morphology of the pharyngeal arches^{60,61}.”

The authors further analyze the role of RA and Fgf8 during early development steps. The authors conclude in ms line 133 that from day 18 to day 28 no further increase in FOXN1 expression occurred, however the referenced Suppl Fig 1d showing relative expression to

day 18 seems to show 10 fold higher expression values? Showing D18 and D28 results next to each other would be helpful to read and interpret the data.

We apologize for the unclear wording and data presentation regarding Supplementary Fig. 1d. We meant that supplementation of RA and FGF8 beyond day 18 does not lead to increased *FOXN1* compared to only base medium without additional factors, so there is no benefit to continuing to add them after day 18. *FOXN1* automatically increases from day 18 to day 28 without any supplementation, likely due to the self-directed differentiation once lineage fates are determined. To make this clearer, we have added the D18 results next to the D28 results in Supplementary Fig. 1d as suggested. We have also added the statistical analysis demonstrating *FOXN1* on day 28 is the same whether RA or FGF8 supplementation continues beyond day 18 or not, showing clearly which groups are being compared. To reflect this, we have amended the wording as follows:

- “When RA or FGF8 supplementation was extended until day 28, no further increase in *FOXN1* expression was seen compared to no supplementation after day 18, [...]”

Introducing the gene regulatory network analysis and its purpose would be beneficial, similarly the term regulon (ms line 292-293).

We have added the definition of the term “regulon” to this line as follows:

- “[...], we performed gene regulatory network (GRN) analysis and identified regulons – groups of genes that consist of a TF and its direct-binding targets – that were highly active in each cluster.”

We have also added an explanation of gene regulatory network analysis and its purpose to the Methods section under the subheading “**Differential expression and GRN analysis**” as follows:

- “GRNs are computational models of TFs regulating each other and their target genes, representing their interactions as networks. GRNs thereby allow the inference of transcriptional states and cell identities.”

The authors might explore a different display version for the results of the ligand-receptor interaction analysis (ms line 313-315, Suppl Fig. 6h). Which ligand-receptor pairs were identified and how many per population?

To clarify this, we have added the number of identified ligand-receptor pairs as black bars above each population in Supplementary Fig. 9a (previously Supplementary Fig. 6h). Since there are hundreds of pairs across the different populations, it would be difficult to display all of them in the figure. We have therefore made a comprehensive list of them in the Source Data file, which shows all pairs with p-value < 0.05, ranked by their interaction

score.

In the discussion the authors state in ms line 441-442 “our results indicate that early cTEC-like cells arise before mTEC lineage commitment and may still have multilineage potential until the downregulation of FOXP1 and cTEC markers”, in addition to Ref 30 the authors should thoroughly cite and discuss previous work done on TEC progenitors and cTEC and mTEC origin. This has been a field of intensive study with opposing results and discussions in the field. Hence the manuscript would benefit from an embedding into those studies. Also, that mTECs derive from cTECs has been proposed and studied before and should be cited.

To better connect our manuscript to existing studies on the origin of cTECs and mTECs, we have added extensive citations of these works to the Discussion section, where we now more thoroughly discuss our results in the context of their findings in the following paragraph:

- “Overall, these data suggest a default pathway to cTECs, from which mTECs sequentially diverge over time during embryonic development, consistent with the asymmetrical serial progression model proposed in mice^{3,115,139}. Previous reports have indicated that this serial progression may be exclusive to the embryonic period, given that in the postnatal murine thymus, the mTEC compartment is mostly maintained by unipotent mTEC-restricted stem cells, which arise from *Psmb11*-expressing progenitors only during early development^{25,26}. While lineage tracing studies have demonstrated the bipotent potential of embryonic cTEC-like cells^{22–24}, adult bipotent progenitors have been detected in various subsets including $Ly51^+MHCII^{high}Plet1^+$ and $Ly51^{low}MHCII^{low}Sca-1^{high}\alpha6\text{-integrin}^{high}$ in mice^{140,141}. Recently, it has also been shown that embryonic but not postnatal $Ccl21^+$ cells are capable of converting into both cTECs and $Aire^+$ mTECs¹⁴², and embryonic $Krt19^+$ and $Sox9^+$ subsets may act as mTEC-biased progenitors^{14,143}. Together, these murine studies indicate multiple possible pathways toward the cTEC and mTEC fates, with differing progenitor identities and levels of plasticity between the fetal and postnatal periods, although whether the same developmental dynamics are present in the human thymus, where certain markers such as *PLET1* are not expressed¹⁴⁴, remains unclear. As iPSC-based systems likely reflect the embryonic rather than the postnatal state, our iTECs offer a unique look into early human thymus development that postnatal thymic epithelial stem cells or TEC lines cannot provide. Indeed, our single cell profiling allowed us to see developmental shifts in the TEP cluster, with RNA velocity further hinting at plasticity between closely related TEC lineages. While the heterogeneity of the terminally differentiated mimetic population is not yet fully

represented in our system, with most mimetic mTEC-like cells by day 133 having a secretory or keratinized phenotype, and all likely post-*AIRE* cells in thymocyte co-culture assuming a neuronal or neuroendocrine phenotype, further improvements in modeling the lymphostromal crosstalk of the thymus *in vitro* may yet yield other subtypes of mimetic mTECs and allow further exploration of their origins in humans.”

Point-by-point response to reviewer comments

Reviewer comments are shown in black; our responses are shown in blue.

Reviewer #1 (Remarks to the Author):

In this revised version, the authors have addressed most of the major concerns raised in the review, primarily through new experimental data. These additions have helped clarify aspects of the manuscript that were previously ambiguous. Furthermore, the authors provided generally well-reasoned and convincing responses to the comments. As a result, the manuscript has been meaningfully improved and now presents a more solid and coherent body of work. The revised study is expected to contribute to the field of iPSC-based in vitro thymic epithelial cell modeling by offering a useful new model that others can build upon.

We thank Reviewer #1 for their insightful feedback, which strengthened our manuscript.

Reviewer #3 (Remarks to the Author):

I wish to clarify that I'm not an original reviewer but that I rather mediate for the missing Reviewer 2 at this stage of the manuscript evaluation process (revision stage).

I have looked at the comments and remarks that were made by both reviewers and it appeared that both reviewers showed enthusiasm for the work already upon initial submission. One particular issue that was raised by both reviewers was the lack of AIRE expressing TECs in this newly designed methodology that aims to generate various human TEC populations from human iPSCs. I agree with both reviewers that this is indeed an important point (but also that this work even without this would provide a substantial advancement for the field as also mentioned). The authors have now addressed this by performing new coculture experiments with thymocytes to show that AIRE expressing cells can be detected in their iPSC-derived TECs, in line with the knowledge that differentiating thymocytes indeed are needed to induce TEC maturation into AIRE expressing subsets through RANK-RANKL interactions. These new results, which are now described in a whole new section in the revised manuscript, not only address these concerns but also significantly strengthen the paper and further solidify the proposed methodology to generate human TECs from iPSCs.

In addition to this, I believe that the authors have adequately addressed the other points that were raised by Reviewer 2, including incorporation of relevant literature, explanation of the role of various compounds that are used in the protocol and some other requested clarifications (terminology and figure improvement).

As such, I believe that the authors have adequately revised their manuscript.

We thank Reviewer #3 for their evaluation of our revised manuscript and their input on the specific points raised by Reviewer #2.

No further changes to the manuscript were made based on these comments.